# Inability to switch from ARID1A-BAF to ARID1B-BAF impairs exit from pluripotency and commitment towards neural crest formation in *ARID1B*-related neurodevelopmental disorders

Luca Pagliaroli [1,9], Patrizia Porazzi[2,9], Alyxandra T. Curtis[1], Chiara Scopa[1], Harald M. M. Mikkers[3], Christian Freund[4], Lucia Daxinger [5], Sandra Deliard[6], Sarah A. Welsh[6], Sarah Offley [6], Connor A. Ott [1], Bruno Calabretta[2], Samantha A. Brugmann[7], Gijs W. E. Santen[8,10] & Marco Trizzino [1,10 ✉]

Subunit switches in the BAF chromatin remodeler are essential during development. *ARID1B* and its paralog *ARID1A* encode for mutually exclusive BAF subunits. De novo *ARID1B* haploinsufficient mutations cause neurodevelopmental disorders, including Coffin-Siris syndrome, which is characterized by neurological and craniofacial features. Here, we leveraged *ARID1B*$^{+/-}$ Coffin-Siris patient-derived iPSCs and modeled cranial neural crest cell (CNCC) formation. We discovered that ARID1B is active only during the first stage of this process, coinciding with neuroectoderm specification, where it is part of a lineage-specific BAF configuration (ARID1B-BAF). ARID1B-BAF regulates exit from pluripotency and lineage commitment by attenuating thousands of enhancers and genes of the *NANOG* and *SOX2* networks. In iPSCs, these enhancers are maintained active by ARID1A-containing BAF. At the onset of differentiation, cells transition from ARID1A- to ARID1B-BAF, eliciting attenuation of the NANOG/SOX2 networks and triggering pluripotency exit. Coffin-Siris patient cells fail to perform the ARID1A/ARID1B switch, and maintain ARID1A-BAF at the pluripotency enhancers throughout all stages of CNCC formation. This leads to persistent NANOG/SOX2 activity which impairs CNCC formation. Despite showing the typical neural crest signature (TFAP2A/SOX9-positive), *ARID1B*-haploinsufficient CNCCs are also aberrantly NANOG-positive. These findings suggest a connection between *ARID1B* mutations, neuroectoderm specification and a pathogenic mechanism for Coffin-Siris syndrome.

[1] Department of Biochemistry and Molecular Biology, Sidney Kimmel Medical College, Thomas Jefferson University, Philadelphia, PA, USA. [2] Department of Cancer Biology, Sidney Kimmel Medical College, Thomas Jefferson University, Philadelphia, PA, USA. [3] Department of Cell & Chemical Biology, Leiden University Medical Center, Leiden, The Netherlands. [4] LUMC hiPSC Hotel, Dept. Anatomy & Embryology, Leiden University Medical Center, Leiden, The Netherlands. [5] Department of Human Genetics, Leiden University Medical Center (LUMC), Leiden 2300 RC, The Netherlands. [6] Gene Expression and Regulation Program, The Wistar Institute, Philadelphia, PA, USA. [7] Divisions of Developmental Biology and Plastic Surgery, Department of Pediatrics at Cincinnati Children's Hospital Medical Center, Cincinnati, OH, USA. [8] Department of Clinical Genetics, Leiden University Medical Center, Leiden, The Netherlands. [9] These authors contributed equally: Luca Pagliaroli, Patrizia Porazzi. [10] These authors jointly supervised: Gijs W. E. Santen, and Marco Trizzino. ✉email: marco.trizzino@jefferson.edu

Cell fate commitment is a complex process that requires timely regulation of developmental genes. This phenomenon is mediated by the concerted activity of transcription factors and chromatin regulators that modulate the interaction between *cis*-regulatory elements (enhancers, promoters) and RNA Polymerase II to promote gene expression. In this framework, a key role is played by the Brg1/Brm associated factor (BAF) chromatin-remodeling complex. BAF leverages ATP to modulate nucleosome positioning and chromatin accessibility genome-wide[1]. Different configurations of BAF, with context-specific functions, have been described, and switches between subunits have been reported to be linked to specific developmental stages[2,3]. All known canonical BAF configurations require the presence of a subunit containing an AT-rich DNA binding domain (ARID). Namely, in the BAF complex, this function is carried out by two mutually exclusive subunits: ARID1A and ARID1B[4–6]. Previous studies in mouse embryonic stem cells (mESCs) have identified an ESC-specific configuration of BAF which regulates pluripotency and self-renewal of embryonic stem cells (esBAF)[4–6]. Importantly, the esBAF predominantly incorporates ARID1A and very rarely ARID1B (relative ARID1B abundance in the esBAF was quantified as ~0 by Ho et al.[6]). One of these studies also identified a non-canonical version of BAF (gBAF), which did not contain an ARID subunit and was involved in pluripotency maintenance of mESCs[4].

De novo haploinsufficient mutations in the *ARID1B* gene cause a spectrum of neurodevelopmental disorders, ranging from Coffin-Siris syndrome to non-syndromic intellectual disability (ID)[7–12]. Coffin-Siris syndrome is associated with ID, specific craniofacial features, growth impairment, feeding difficulties, and congenital anomalies such as heart and kidney defects[13]. Although other BAF components may also be mutated in this syndrome, ~75% of the mutations are in *ARID1B*[11,14,15]. In addition to Coffin-Siris, genome-wide sequencing in unselected cohorts of patients with ID shows that *ARID1B* is always in the top-5 of causative genes, explaining ~1% of all ID cases[9,16]. Whereas several studies utilizing murine models recapitulate the neurological phenotypes typical of the *ARID1B*-associated syndromes[17–20], the molecular function of ARID1B in cell fate commitment during human development is still poorly understood.

Several hallmark features of *ARID1B*-haploinsufficient individuals, including severe craniofacial, cardiac and digestive system abnormalities, suggest impaired neural crest cell migration as a pathological etiology[12]. Further, *ARID1B* is one of the most commonly mutated genes in neuroblastoma, a pediatric tumor of neural crest origin[21]. Thus, neural crest formation, migration, and differentiation represent suitable models to study the consequences of *ARID1B* mutations. To specifically address the molecular consequences of *ARID1B*-haploinsufficient mutations in neural crest formation and development, we reprogrammed skin fibroblasts of two unrelated *ARID1B+/−* Coffin-Siris patients into induced Pluripotent Stem Cells (iPSCs) and used these patient-derived iPSCs to specifically model formation of cranial neural crest cell (CNCC), a multipotent cell population that forms through a neuroectodermal sphere intermediate that eventually gives rise to migratory CNCCs.

Herein, we report the discovery of a lineage-specific BAF configuration, containing ARID1B, SMARCA4, and additional subunits (ARID1B-BAF). In line with findings indicating that the esBAF and the gBAF do not contain ARID1B[4–6], we demonstrate that *ARID1B* mutations do not affect self-renewal and pluripotency of human iPSCs, as pluripotency is conveyed via binding of an ARID1A-containing BAF to pluripotency-associated enhancers of the SOX2 and NANOG networks. On the other hand, we show that ARID1B-BAF is required for lineage specification and exit from pluripotency. In fact, ARID1B-BAF is only transiently active during the early stages of iPSC-to-CNCC differentiation, and specifically during the formation of the neuroectodermal spheres, where it replaces ARID1A-BAF at the SOX2/NANOG enhancers and elicits their repression.

Importantly, we demonstrate that *ARID1B+/−* cells from Coffin-Siris patients are unable to switch from ARID1A-BAF to ARID1B-BAF at the onset of neuroectoderm formation, and instead maintain ARID1A-BAF at pluripotency enhancers throughout the entire differentiation process. Failure to replace ARID1A with ARID1B leads to defective exit from pluripotency and impaired cranial neural crest formation. These findings provide evidence for a direct connection between *ARID1B* mutations and a pathogenic mechanism for ARID1B-associated neurodevelopmental syndromes.

## Results

**Coffin-Siris patient-derived iPSCs are pluripotent and proliferate normally.** To investigate the function of ARID1B in craniofacial development, we obtained skin fibroblasts from two unrelated *ARID1B+/−* Coffin-Siris Syndrome patients (hereafter, Patient 19 and Patient 26; Fig. 1a, b), one male and one female, both carrying previously identified de novo *ARID1B* mutations. In detail, Patient 19 presented a non-sense mutation (c.3223 C > T;p.Arg1075*; Fig. 1b), whereas Patient 26 had a frameshift mutation (c.2598del;Tyr867Thrfs*47; Fig. 1b)[10,14]. In both cases, a premature STOP codon was generated (Fig. 1b).

The fibroblasts were reprogrammed into iPSCs by the LUMC hiPSC Hotel (Leiden University). Patient-derived iPSCs exhibited regular morphology (Fig. 1c) and expressed pluripotency genes, as shown by both immunofluorescence (Fig. 1d) and RT-qPCR (Fig. 1e). Further, patient-derived iPSCs grew at the same rate as an *ARID1B+/+* control line (Control line-1; Fig. 1f).

Importantly, the aberrant STOP codon introduced by the mutations was located either upstream (Patient 26) or inside (Patient 19) the AT-rich interactive domain (ARID) (Fig. 1b), which is required for ARID1B's interaction with chromatin[22]. Moreover, in both patients, the new STOP codon was localized upstream of the nuclear localization signal (NLS, Fig. 1b), suggesting that the gene product arising from the mutated allele would not be able to reach the nucleus or access the chromatin, even in the unlikely case that the transcript escaped non-sense mediated mRNA decay[23]. To test this, we performed cellular fractionation in patient and control iPSCs and conducted an ARID1B western blot on the chromatin fraction with an antibody raised against a peptide in the N-terminus of ARID1B, upstream of the mutated regions (sc-32762). Consistent with our hypothesis, the immunoblot on the chromatin fraction revealed a significantly reduced amount of ARID1B protein in both patient samples relative to the control *ARID1B+/+* iPSC line (Control line-1; Supplementary Fig. 1a). ARID1B was reduced by ~60% and 80% in Patients 19 and 26, respectively. ARID1B was not detected in the cytoplasmic or nuclear fraction of any cell lines (Supplementary Fig S1a). Thus, *ARID1B*-haploinsufficient iPSCs remained pluripotent, did not exhibit any growth defects, but did display significantly less chromatin-bound ARID1B than control iPSCs.

**CNCC formation is impaired in Coffin-Siris patient-derived iPSCs.** Utilizing published methods for iPSC-to-CNCC differentiation[24] we generated CNCCs from control iPSCs, in 14 days (Fig. 2a, b). A time-course western blot conducted during the differentiation of Control line-1 revealed that ARID1B protein was robustly expressed during the first week of differentiation, peaking between days 5 and 7, with some expected variability between biological replicates (Fig. 2c; see Fig. 7 for a second replicate). After day 7; however, ARID1B protein was markedly

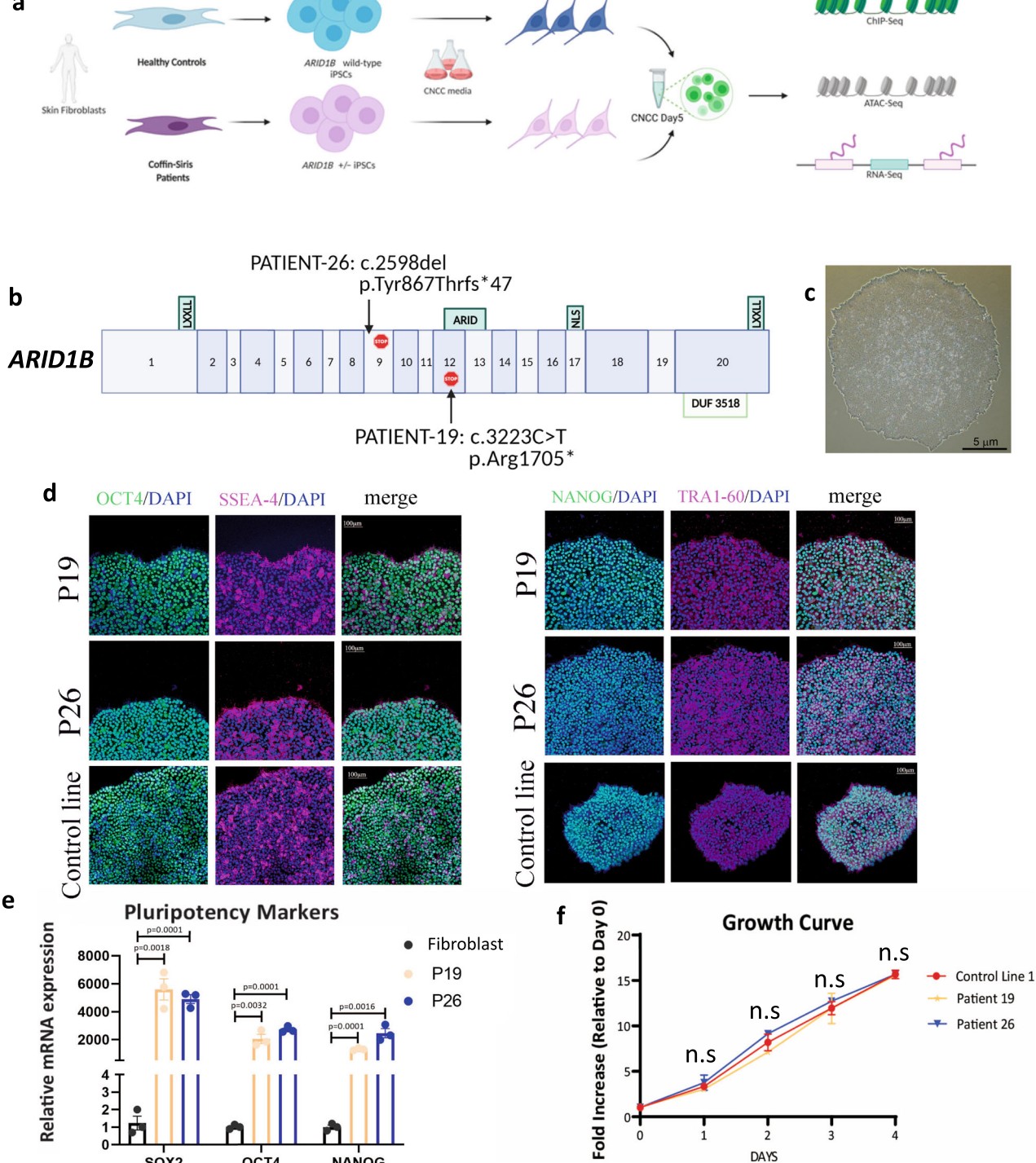

**Fig. 1 iPSCs derived from Coffin-Siris patients are pluripotent and proliferate normally. a** Study system: iPSCs were derived from skin fibroblasts of two unrelated Coffin-Siris patients. The iPSCs were used in this study to generate Cranial Neural Crest Cells (CNCCs) and perform genomic experiments to investigate the effect of *ARID1B* mutations. Panel created by LP. **b** Graphical illustration of the *ARID1B*-haploinsufficient mutations affecting the two studied patients. The numbers in the gene model refer to *ARID1B*'s isoform NM_020732.3. **c** iPSC colony derived from Patient 19. **d, e** Immunofluorescence and rt-qPCR quantifying the expression of the key pluripotency markers in iPSCs derived from Control and Patient lines. *P* values are from two-sided Student's *t* test. *N* = 3 biologically independent samples. Error bars display standard errors. **f** Growth curve comparing an *ARID1B*-wt iPSC Line (CTRL-1) with the two patient lines. *P* values from two-sided Student's *t* test are >0.05 (not significant) in all comparisons. *N* = 3 biologically independent experiments. Error bars display standard errors.

downregulated (Fig. 2c). The window of robust ARID1B expression (day 0 to day 7) coincided with the differentiation of the iPSCs into neuroectodermal spheres, suggesting that this BAF subunit may have a role in neuroectoderm specification.

Next, we induced CNCC differentiation in two Coffin-Siris patient lines and compared them with Control iPSC-derived CNCCs using flow cytometry to measure multiple pluripotency (SSEA4, TRA-1-60-R) and CNCC (CD10, CD99) surface markers. Cells were sampled at

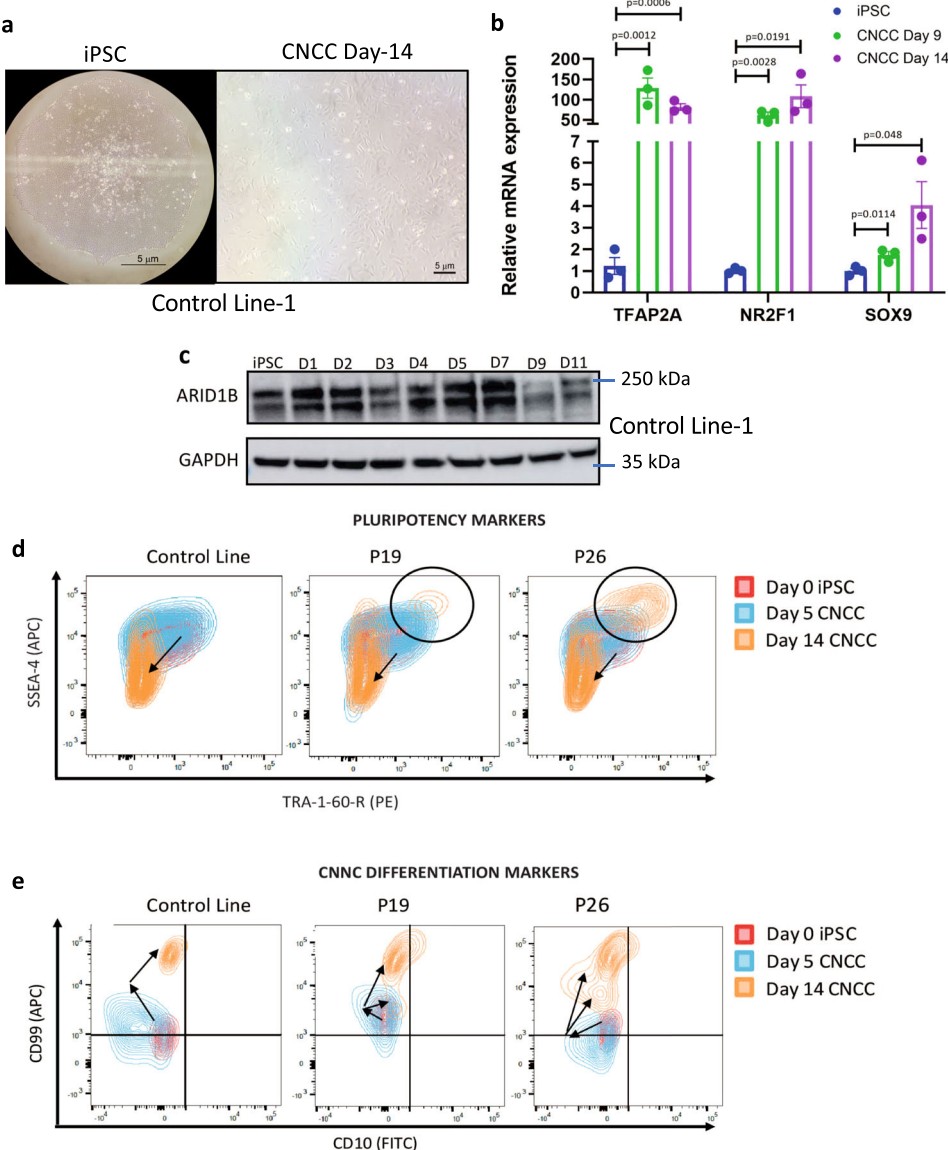

**Fig. 2 CNCC differentiation is impaired in the patient cells. a, b** CNCC differentiation was optimized using an *ARID1B*-wt control line. After 14 days, the cells exhibited the classic CNCC morphology and expressed the CNCC markers. *P* values are from two-sided Student's *t* test. *N* = 3 biologically independent samples. Error bars display standard errors. **c** Time-course immunoblot conducted using Control Line-1 during CNCC differentiation shows that ARID1B is active in the first 7 days of the differentiation, with a peak of activity between day 5 and day 7. The ARID1B protein level strongly decreases after day 7. The experiment was repeated twice (the second replicate is shown in Fig. 7). Marker bars display kDa. **d, e** Flow cytometry quantifying the expression of surface markers for pluripotency and CNCC differentiation in Control Line-1 and in the two patient lines. A large cell population is still pluripotent in both patients after 14 days (**d**). The patient lines also show reduced expression of CNCC surface markers after 14 days of differentiation relative to an *ARID1B*-wt control line at the same time point (**e**).

day-zero (iPSCs), day 5 (neuroectoderm), and day 14 (CNCC). Notably, CNCC formation was impaired in both patient-derived lines, as evident by a sizable cell population that was double-positive for pluripotency surface markers even after 14 days of differentiation (Fig. 2d; Supplementary Fig. 2). This double-positive population comprised 4.5% and 19.5% of the cells in the two patient lines, respectively (Fig. 2d). In line with this, a large fraction of patient cells showed significantly lower expression of CNCC surface markers even after 14 days of differentiation relative to the control line (Fig. 2e).

To further characterize patient-derived CNCCs, we performed immunofluorescence for pluripotency (OCT4, NANOG) and neural crest (SOX9) markers, in control and patient lines at day 14. Interestingly, CNCCs derived from patient iPSCs displayed a gene expression signature distinct from that of control CNCCs

(Fig. 3). In fact, nearly all patient-derived CNCCs were positive for SOX9, NANOG (Patients 19 and 26), and OCT4 (Patient 26 only; Fig. 3). This was in sharp contrast to the control line, in which SOX9-positive cells were almost all OCT4- and NANOG-negative (Fig. 3; See Supplementary Fig. 3 for quantifications and *p* values). Together, these data suggested that while *ARID1B* is dispensable for pluripotency, haploinsufficiency of this gene severely impaired CNCC formation, resulting in CNCCs expressing key pluripotency genes.

**Chromatin accessibility is dysregulated in differentiating Coffin-Siris patient-derived lines**. We used next-generation sequencing to investigate why *ARID1B*-haploinsufficient Coffin-

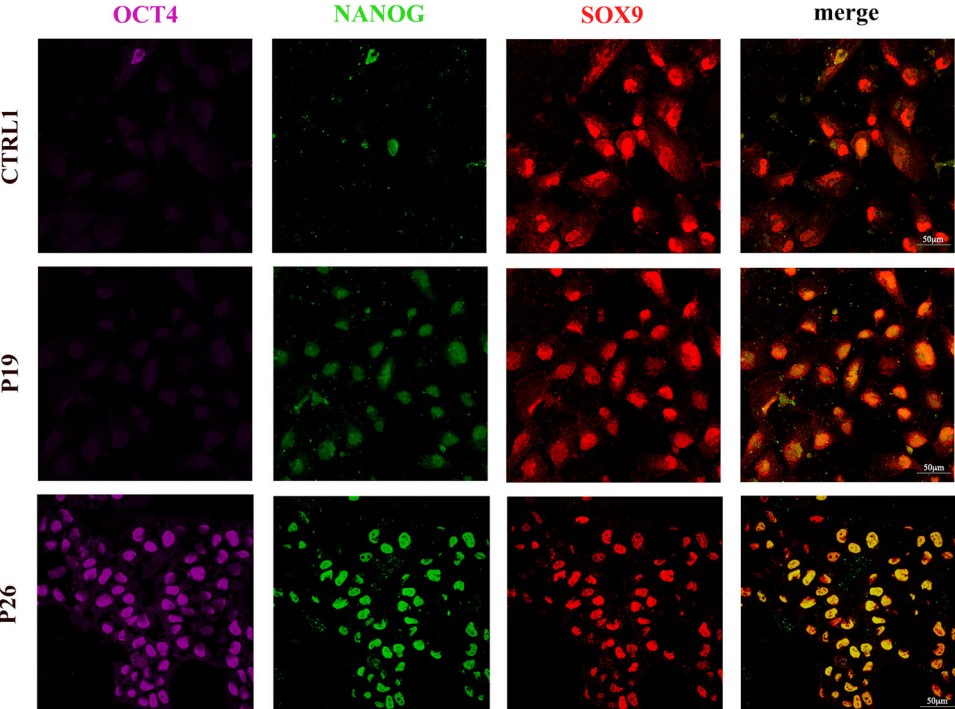

**Fig. 3 Aberrant NANOG and OCT4 activity in the patient-derived CNCC.** Immunofluorescence for SOX9, NANOG, and OCT4 was performed on day 14 of iPSC-to-CNCC differentiation in Control Line-1, Patient 19, and Patient 26 cells.

Siris iPSCs did not successfully differentiate into CNCCs. Given that ARID1B protein levels in control cells reach a peak between days 5 and 7, samples were taken on day 5 to perform genomic analyses. Experiments were conducted with two biological replicates per condition (two control lines, two patient lines). For each condition, a male and a female were included to avoid sex-specific confounding effects. Technical replicates were also used for each biological replicate. To avoid batch effects, all the biological replicates and conditions were processed together.

Since ARID1B is a component of the BAF chromatin-remodeling complex, we profiled chromatin accessibility with ATAC-seq. Overall, at day 5, 29,758 ATAC-seq peaks were identified across all replicates and conditions (patients and controls; false discovery rate (FDR) < 0.05; Fig. 4a). Conversely, 5540 peaks were specific to the patient iPSCs (i.e., replicated in all patient's iPSC replicates and not detected in any of the controls; hereafter patient-specific ATAC-seq regions; Fig. 4a, b; Supplementary Data 1). Finally, only 578 peaks were specific to the controls (hereafter control-specific ATAC-seq regions; Fig. 4a, c; Supplementary Data 1). We, therefore, focused on the 5540 patient-specific ATAC-seq regions because they represented 91% (5540/6118) of all regions with differential chromatin accessibility between patient and control lines.

On day 0, ATAC-seq performed in iPSCs revealed that the 5540 regions were highly accessible, with no significant differences between patient and control lines (Supplementary Fig. 4a). By day 5, this dramatically shifted and 5511 of the 5540 regions (99.4%) were called peaks exclusive to the patient lines. These data suggested that these were regions highly accessible in iPSCs and repressed by day 5 of iPSC-to-CNCC differentiation. Such repression is impaired by *ARID1B* haploinsufficiency, indicating that chromatin accessibility in the 5540 genomic sites may be directly regulated by an ARID1B-containing BAF during exit from pluripotency and neuroectoderm specification. Thus, we investigated ARID1B's binding at these regions both in day 0 (iPSCs) and in day 5 cells. On day 0, these regions were not bound by ARID1B (Supplementary Fig. 4b). This was expected, given that that ARID1B is not found in the gBAF,

whereas ARID1A is the predominant ARID1 in the esBAF[4–6], and likely explains why differences in chromatin accessibility at these regions between patient and control lines at day 0 was not observed.

On the other hand, at the day 5, the 5540 regions were bound by ARID1B in both control lines (6486 ARID1B peaks detected on average in the two CTRL lines), while the binding was almost entirely lost in both patient lines (Fig. 4d; only 750 peaks detected on average in the *ARID1B*-deficient patient lines). This loss of binding correlated with a lack of chromatin repression in both patient lines. Together, these findings indicated that gain of ARID1B binding at these 5540 genomic sites in the early stages of iPSC-to-CNCC differentiation may be required for repression of the pluripotency program.

**The ARID1B-BAF attenuates thousands of enhancers at the onset of CNCC differentiation**. To determine the nature of the 5540 genomic regions, we associated a gene to each region based on the distance from the nearest transcription start site (TSS). Overall, 87.5% of the ARID1B ChIP-seq peaks were located >10 Kbs from the nearest TSS and may represent putative enhancers, whereas the remaining 12.5% are likely promoters. ChIP-seq time-course for H3K27ac in control cells revealed that many of these regulatory regions were enriched for H3K27ac in iPSCs, progressively lost this enrichment during differentiation, and by day 5 had very reduced H3K27ac signal (Fig. 4e). The gradual decrease in H3K27ac mirrored the steady increase in ARID1B expression detected during the early stages of iPSC-to-CNCC differentiation, and specifically during the formation of the neuroectodermal spheres (Fig. 2c). Consistent with this, the differentiating cells from both patients had significantly higher levels of H3K27ac in these regions, relative to the two control lines at day 5 (Wilcoxon's Rank-Sum Test $p < 2.2 \times 10^{-16}$ in all the patient vs control pairwise comparisons; Fig. 4f, g).

We further investigated the fate of these regions over the course of iPSC-to-CNCC differentiation. We performed ChIP-seq

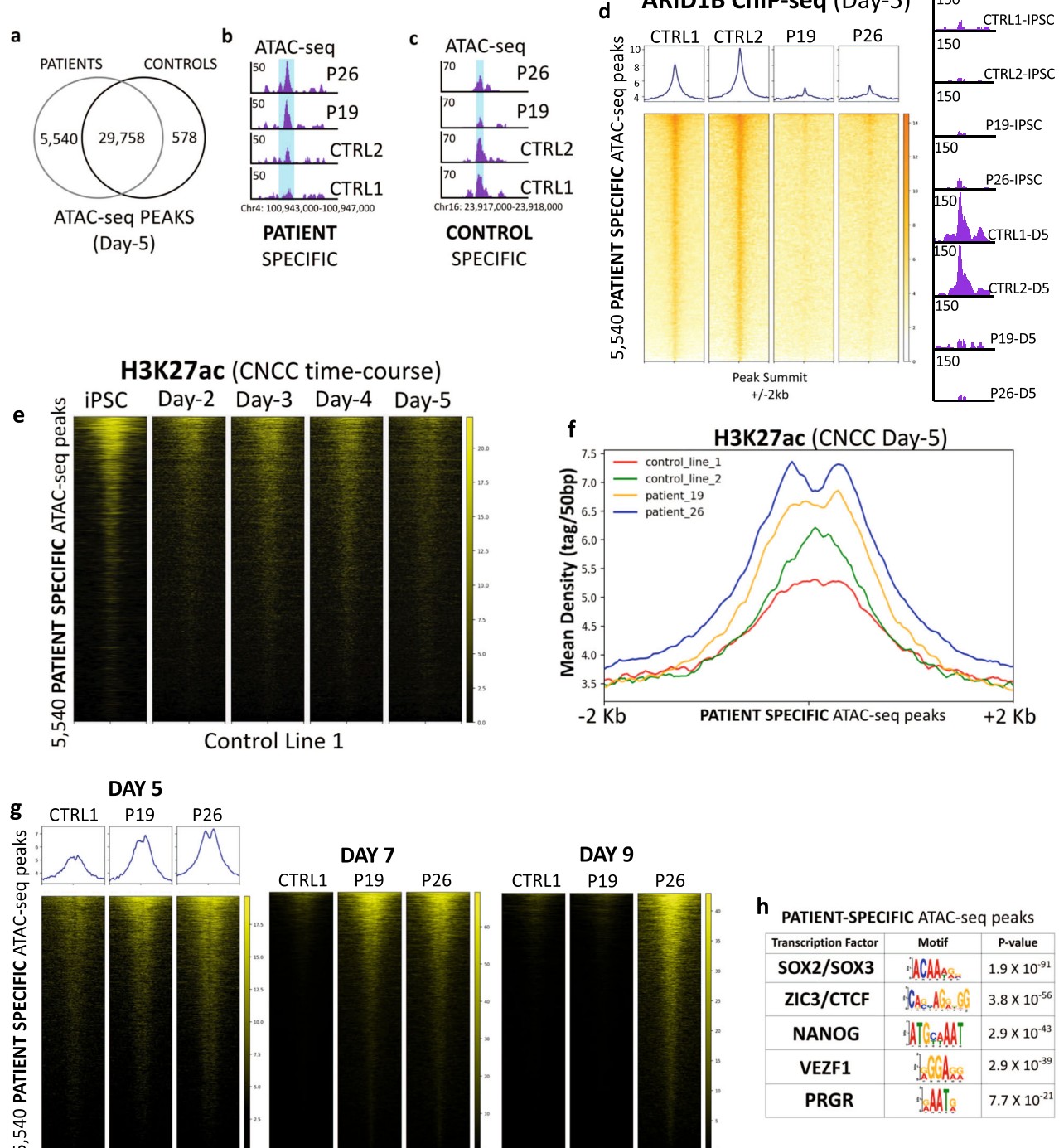

**Fig. 4 Chromatin remodeling at pluripotency enhancers is dysregulated in the patient cells. a** At CNCC day 5, 29,578 ATAC-seq peaks are shared between patient and control lines. In all, 5540 peaks are specific to the patients. 578 peaks are specific to the controls. **b** UCSC Genome Browser example of a patient-specific ATAC-seq peak. **c** UCSC Genome Browser example of a CONTROL-SPECIFIC ATAC-seq peak. **d** ARID1B ChIP-seq heatmaps (Four lines; CNCC Day 5) centered on the patient-specific ATAC-seq peaks; the UCSC screenshot shows ARID1B binding at a patient-specific ATAC-seq peak (chr10:129,590,000–129,595,000) in the control and patient lines at iPSC stage and at CNCC Day 5. **e** Spike-in normalized heatmaps of H3K27ac ChIP-seq time-course at the 5540 patient-specific ATAC-seq peaks (Control Line-1). **f** Spike-in normalized H3K27ac ChIP-seq average profiles centered on the patient-specific ATAC-seq regions (CNCC Day 5). Statistical significance was assessed with Wilcoxon's Rank-Sum test ($p < 2.2 \times 10^{-16}$ in all the patient vs control comparisons). **g** Spike-in normalized heatmaps of H3K27ac ChIP-seq signal at Days 5, 7, and 9 at the 5540 patient-specific ATAC-seq peaks (Control Line-1, Patient 19, Patient 26). **h** Motif analysis at the patient-specific ATAC-seq regions revealed enrichment for the binding motif of multiple pluripotency factors. *P* values are calculated by Meme-ChIP as *E* values. The *E* value is the motif *P* value (computed by Fisher's Exact Test) times the number of candidate motifs tested.

for H3K27ac in CTRL line-1, P19, and P26 on days 5, 7, and 9 (peak numbers ranged from 96,600 on day 5 to 153,134 on day 9).

We found that the patient-specific cis-regulatory elements were inactive in the control line at day 7, but still largely active in the patient lines at the same time point (Fig. 4g). Interestingly, on day 9 of differentiation, these regions were inactivated (i.e., no H3K27ac signal) in Patient 19 line, while remaining active (i.e., persistent H3K27ac signal) in the Patient 26 line (Fig. 4g).

Based on the high H3K27ac signal that the 5540 patient-specific regions display at the iPSC stage (Fig. 4e), we surmised that these sites could represent cis-regulatory elements important for pluripotency. In line with this hypothesis, DNA-motif analysis on the 5540 regions revealed that they were enriched for the binding sites of multiple pluripotency factors, including SOX2 and NANOG (Fig. 4h; Supplementary Data 2).

To ensure that the molecular phenotypes observed were directly caused by the *ARID1B* mutations, and not by co-occurring mutations in other genes coincidentally shared by both (unrelated) patients we employed shRNAs to knock-down ARID1B in the Control Line-1. We were able to obtain a partial knock-down of ARID1B at the iPSC stage (shRNA-1; Supplementary Fig. 4c), which represented a suitable model for *ARID1B* haploinsufficiency. ARID1B-KD iPSCs were put through the CNCC differentiation protocol, collected at day 5, and profiled via ATAC-seq and ChIP-seq for H3K27ac. Notably, both sequencing experiments perfectly recapitulated our findings in the patient-derived lines. Upon ARID1B-KD, we detected significantly increased chromatin accessibility and H3K27ac signal in the 5540 patients-specific regions relative to the same iPSC line transduced with a control shRNA (Wilcoxon's rank-sum test $p < 2.2 \times 10^{-16}$; Supplementary Fig. 4d, e).

Together, these data indicate that ARID1B-BAF modulates the chromatin accessibility of a specific set of ~4900 pluripotency enhancers and ~600 promoters that are highly active in iPSCs, moderately active at the onset of neuroectoderm formation, and inactive by day 7 and for the remaining course of CNCC formation (Fig. 4e). These data suggested that impaired attenuation of these cis-regulatory elements in the *ARID1B*-haploinsufficient cells, subsequently hampers the entire differentiation process towards a CNCC fate.

**Pluripotency and exit from pluripotency genes are dysregulated in differentiating patient lines**. Impaired attenuation of ~4900 pluripotency-relevant enhancers and ~600 promoters could have a profound effect on gene expression levels. Indeed, RNA-seq conducted at day 5 identified 2356 differentially expressed genes, 1685 of which were downregulated, and 671 upregulated (FDR < 5%; Fig. 5a). As expected, *ARID1B* was one of the top downregulated genes in patient CNCCs (Fig. 5a). In stark contrast, only 54 genes were identified as differentially expressed when we performed RNA-seq at the iPSC stage (FDR < 5%). This suggested that ARID1B is important for lineage commitment, again mirroring the progressive increase in the ARID1B protein level observed during the early stages of differentiation (Fig. 2c). The small number of differentially expressed genes identified at the iPSC stage was again consistent with the finding that esBAF and gBAF do not include ARID1B[4–6].

Notably, 598/2356 (25.4%) of the genes differentially expressed at day 5 also represented the nearest gene to one of the 5540 pluripotency enhancers and promoters aberrantly active in the Coffin-Siris patient cells at the same time point (Supplementary Data 3). These results suggested that over a quarter of differentially expressed genes were under the direct control of ARID1B-BAF throughout modulation of chromatin accessibility at associated enhancers and promoters. As expected, when we compared these

598 genes against the entire set of 2356 differentially expressed genes, we found that the 598 genes exhibited enrichment for genes upregulated in patient cells (Fisher's Exact Test $p < 0.0001$). Ingenuity-Pathway Analysis on the 598 genes identified five of the top canonical pathways as associated with either pluripotency or exit from pluripotency, as well as Wnt-β catenin signaling pathway[25,26] (Fig. 5b; Supplementary Data 4).

In accordance with the ATAC-seq data, SOX2 was detected among the top upstream regulators (Fig. 5c), and three of the most important pluripotency factors, *NANOG, SOX2,* and *POU5F1* (OCT4) were highly expressed in the patient lines at day 5 (Fig. 5d).

Both the [Role of NANOG in Mammalian Embryonic Stem Cell Pluripotency] and the [PPARα/RXRα Activation] pathways were enriched in the 598 genes (Fig. 5b). Namely, the genes belonging to the former pathway were all upregulated, while those belonging to the latter were downregulated (Fig. 5d). These two pathways caught our attention because they are thought to antagonize each other. More specifically, NANOG blocks the differentiation of pluripotent cells and establishes the pluripotent state during somatic cell reprogramming. Conversely, the PPARα/RXRα pathway is activated at the onset of differentiation to promote exit from pluripotency[27]. The activation of PPARα/RXRα contributes to the repression of the *NANOG* network to allow efficient exit from the undifferentiated stage[27–29]. Consistent with this, PPARα-inhibitors have been employed to improve iPSC reprogramming[27].

These gene expression data were in line with the immuno-fluorescence findings and with the perpetual *NANOG* and *OCT4* expression in patient-derived CNCCs at day 14 of differentiation (Fig. 3).

Finally, we looked for potential overlap between our set of 2356 differentially expressed genes at day 5 of iPSC differentiation toward CNCC and a set of genes identified as differentially expressed in a recent RNA-seq study which compared cerebellar tissue of *ARID1B*$^{+/-}$ and *ARID1B-wt* mice[30]. Notably, 1297 of the 2356 genes (55%) found as differentially expressed in our study were also found as differentially expressed in the KO-mouse model data set. This overlap (55%) was significantly higher than expected by chance (Fisher's exact test $p < 2.2 \times 10^{-16}$) and suggested that the pathways regulated by ARID1B are important for both craniofacial and brain development.

Taken together, our RNA-seq data suggested that differentiating *ARID1B*$^{+/-}$ patient-derived lines exhibited a persistent upregulation of multiple pluripotency factors and associated gene networks, along with downregulation of genes responsible for exit from pluripotency. Dysregulation of these gene networks subsequently impaired neuroectoderm specification and CNCC formation.

**Aberrant SOX2 and NANOG activity in the *ARID1B*-haploinsufficient patient cells**. Our experiments indicate that the *ARID1B*-haploinsufficient cells fail to attenuate thousands of pluripotency enhancers and promoters enriched for SOX2 and NANOG-binding sites (Fig. 4a–h). Further, on day 5 of iPSC-to-CNCC differentiation, the expression of *SOX2* and *NANOG* was significantly higher in the patient-derived cells than in the controls, and the gene regulatory networks associated with these pluripotency factors were also upregulated (Fig. 5b–d). Moreover, both patient-derived CNCCs (day 14) exhibited aberrant expression of NANOG (Fig. 3).

Given these findings, we set out to investigate the binding profile of SOX2 and NANOG in patient and control lines by ChIP-seq at CNCC day 5. Our spike-in normalized SOX2 ChIP-seq revealed that 3284/5540 (59.7%) patient-specific ATAC-seq peaks exhibit significantly higher SOX2 binding in patients

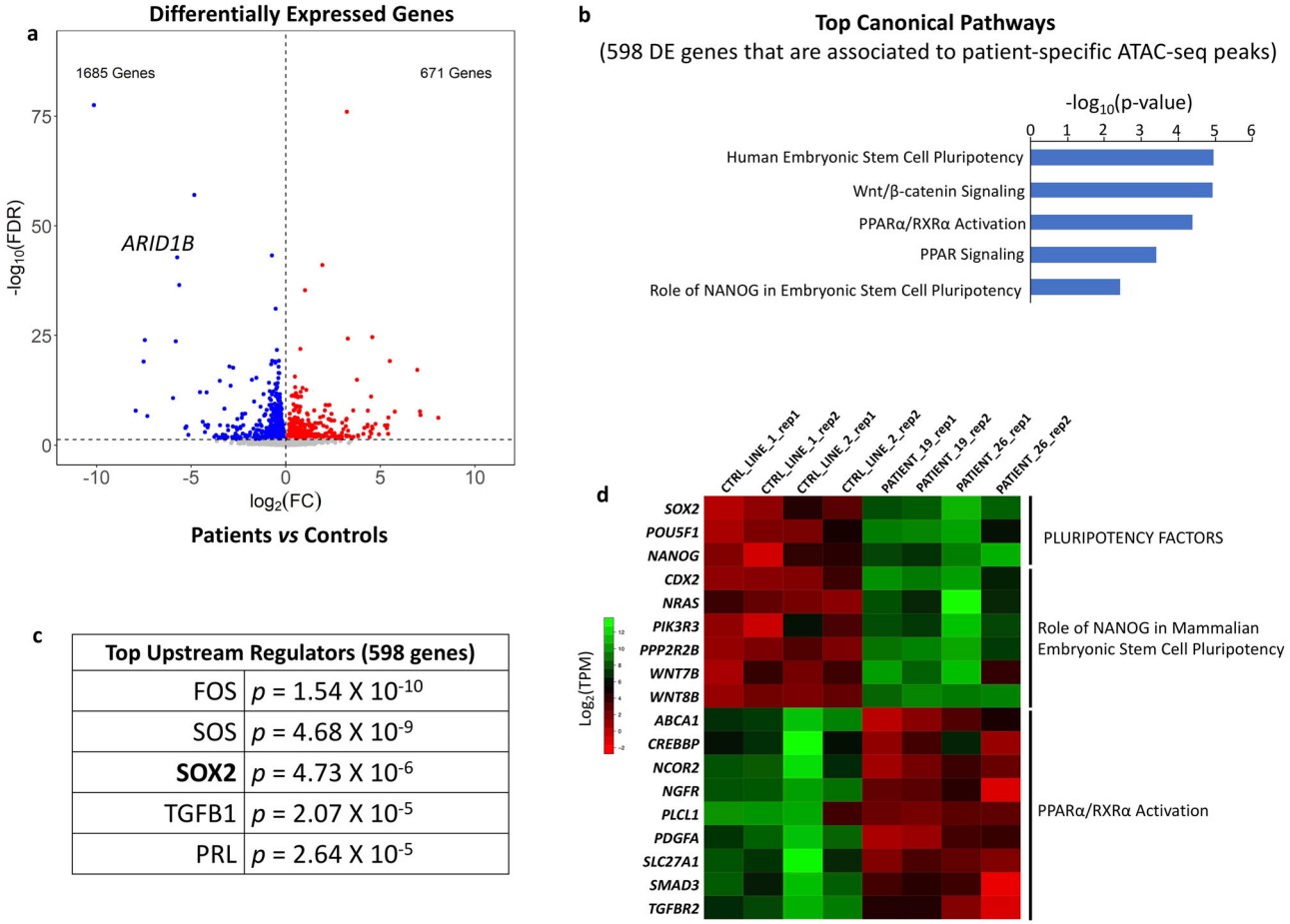

**Fig. 5 Pluripotency and exit from pluripotency genes are dysregulated in differentiating patient CNCCs. a** RNA-seq volcano plot shows the differentially expressed genes between patient and control lines at CNCC Day 5. *ARID1B* is one of the top downregulated genes. **b** Top canonical pathways (IPA analysis) enriched in the set of 598 differentially expressed genes that also represent the closest gene to a patient-specific ATAC-seq peak. **c** Top upstream regulators (IPA analysis) enriched in the same set of 598 genes used for **b**. *P* values are calculated with Fisher's Exact Test, adjusted using False Discovery Rate (Benjamini–Hochberg). **d** RNA-seq heatmap displaying expression patterns at CNCC Day 5 for pluripotency genes, for genes of the NANOG network, and for genes associated to exit from pluripotency (PPARα/RXRα activation pathway).

relative to control lines (Fig. 6a). In line with this, the chromatin at these regions is accessible in the patient lines but not in the control lines (Fig. 6b). SOX2 is a pioneer factor that can bind condensed nucleosomes to open the chromatin and allow binding of other factors[31]. As demonstrated by previous studies in mESCs, SOX2 and other pluripotency pioneer factors (e.g., OCT4) require the BAF complex to perform their pioneer activity[6,31,32]. Our findings indicate that, in control conditions, the ARID1B-BAF complex likely antagonizes the cooperation between other BAF configurations and SOX2, counter-acting the pioneer activity of the latter as soon as cell differentiation is induced. Further, we identified an additional set of 497 SOX2 peaks specific to the patient lines, which did not exhibit changes in chromatin accessibility. Moreover, we also identified 1146 SOX2 peaks exclusive of the control lines (Supplementary Data 5). Importantly, these control-specific SOX2 peaks were located in proximity to genes associated with neural crest differentiation, including *TFAP2A*, *PAX6*, *PAX7*, *WNT4*, *ENO1*, *C8B*, and *SERBP1* among others. These findings are consistent with two recent studies which suggested that SOX2-chromatin interactions are rewired upon differentiation cues[33,34]. Such rewiring appears impaired in *ARID1B*-haploinsufficient cells, which aberrantly maintain SOX2 at pluripotency-associated enhancers, and at the same time fail to reposition this transcription factor at the developmental enhancers.

Next, we profiled NANOG at day 5 of differentiation. For this transcription factor, the spike-in normalized ChIP-seq revealed 4538 peaks unique to the patients (Supplementary Data 6). However, in this case, only 219 (4.8%) of the patient-specific NANOG peaks overlapped a patient-specific ATAC-seq peak. We thus interrogated our ATAC-seq data to determine the state of chromatin accessibility at the 4538 patient-specific NANOG peaks, and overall found no significant changes in accessibility in these regions between the patients and the control lines (Fig. 5c). We note that nearly a quarter of the patient-specific NANOG peaks were found in regions of repressed chromatin (Fig. 6c, e), in line with recent studies which suggested that NANOG can bind repressed chromatin like other pioneer pluripotency factors[35,36].

Despite no changes in chromatin accessibility, the NANOG ChIP-seq signal at the 4538 patient-specific NANOG peaks was significantly higher in the patient than in the control lines (Wilcoxon's rank-sum test $p < 2.2 \times 10^{-16}$ in all the patient vs control pairwise comparisons; Fig. 6d, e). We hypothesized that the increased NANOG binding detected in the patients' cells (Fig. 6d) could reflect increased *NANOG* expression (Fig. 6d). In fact, several elegant studies in embryonic stem cells have demonstrated that changes in *NANOG* dosage mark the transition from an undifferentiated state (high *NANOG*), to a state with differentiation potential (low *NANOG*)[37–39]. Importantly, it has been shown that SOX2 and OCT4 bind a cis-

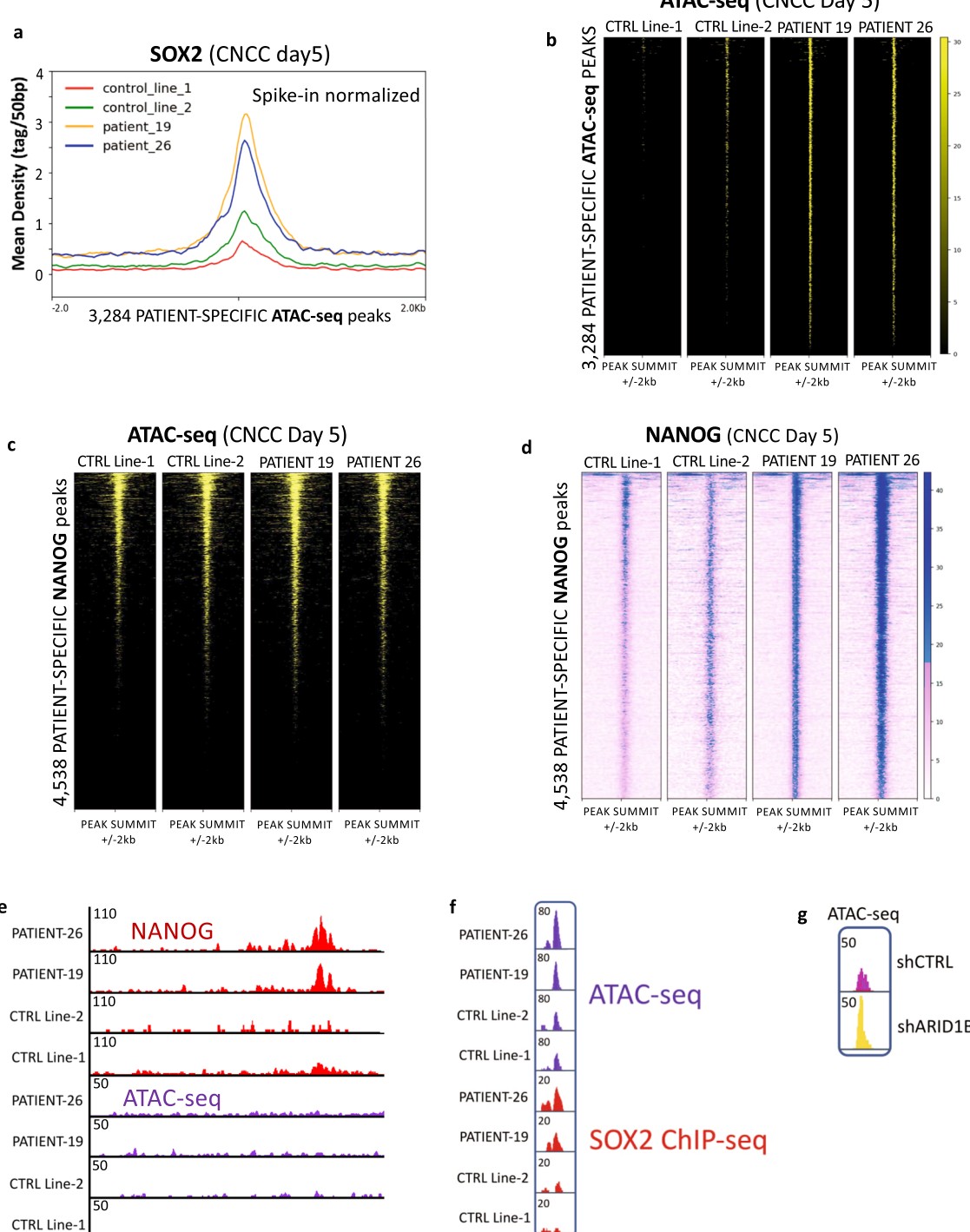

**Fig. 6 Aberrant SOX2 and NANOG activity in the patient cells at CNCC day 5. a** Spike-in normalized SOX2 ChIP-seq average profile for 3284 patient-specific ATAC-seq peaks showing patient-specific SOX2 signal (spike-in normalized; CNCC Day 5). Statistical significance was assessed with Wilcoxon's Rank-Sum test ($p < 2.2 \times 10^{-16}$ in all the patient vs control comparisons). **b** ATAC-seq heatmaps at the 3284 peaks shown in Fig. 5a reveal that these regions display increased chromatin accessibility in the patients relative to the two control lines. **c** ATAC-seq heatmaps at 4538 patient-specific NANOG peaks display no changes in accessibility between patient and control lines. **d** Spike-in normalized NANOG ChIP-seq heatmaps at 4538 patient-specific NANOG peaks (spike-in normalized; CNCC Day 5). **e** Example of patient-specific NANOG peak in a region with no chromatin accessibility (CNCC Day 5). **f** At CNCC Day 5, a cis-regulatory element in the promoter region of *NANOG* is more accessible in the patients than in the control lines. The same element also displays higher SOX2 binding in the patients than in the controls. **g** Knockdown of ARID1B from Control Line-1 also elicits an increase in chromatin accessibility at the cis-regulatory element in the promoter region of *NANOG* (CNCC Day 5).

regulatory element in the promoter region of *NANOG*, likely modulating its expression[40,41]. Thus, we examined this cis-regulatory element in detail. As expected, at day 5 of iPSC-to-CNCC differentiation, the chromatin accessibility at the promoter-proximal element is significantly higher in the two patient lines than in the two controls (Student's *t* test *p* = 0.0065; Fig. 6f). Accordingly, increased chromatin accessibility correlates with increased SOX2 binding on the cis-regulatory element (Fig. 6f), perhaps explaining the higher *NANOG* gene expression reported in patient-derived lines at day 5. At last, our shRNA experiments also confirmed these findings, demonstrating that the knockdown of ARID1B in the Control line-1 line correlates with a sizeable increase in accessibility at the *NANOG* cis-regulatory element (Fig. 6g), thus suggesting that ARID1B-BAF directly modulates *NANOG* expression dosage at the onset of differentiation.

In sum, the *ARID1B*-haploinsufficient lines exhibit the persistent activity of two key pluripotency factors (SOX2, NANOG) in the early and late stages of CNCC formation. The aberrant activity of SOX2 and NANOG likely leads to impaired lineage commitment and inefficient CNCC formation.

**A switch from ARID1A-BAF to ARID1B-BAF is necessary for exit from pluripotency.** We next wanted to elucidate how the BAF complex compensated for *ARID1B* haploinsufficiency. As mentioned above, ARID1A and ARID1B represent the only two subunits of the BAF harboring an ARID domain, which is leveraged by the complex to interact with the chromatin[22]. A third ARID subunit (ARID2) is exclusive to a different configuration of the complex (pBAF). *ARID2* mutations have been shown to cause a neurodevelopmental disorder that does not fully recapitulate the Coffin-Siris syndrome phenotype, although there is some overlap[42]. Compensatory mechanisms between ARID1A and ARID1B were recently demonstrated in ovarian cancer[43]. Thus, we hypothesized that *ARID1B*-haploinsufficient patient-derived cells may compensate for partial loss of ARID1B with ARID1A. To test this, we first assessed ARID1A protein levels in *ARID1B*-wt cells during CNCC formation and found that ARID1A expression was complementary to ARID1B (Fig. 7a). In agreement with the specific composition of the esBAF, which requires ARID1A[5,6], human iPSCs expressed high levels of ARID1A and relatively low ARID1B (Fig. 7a). On the other hand, at the initiation of iPSC-to-CNCC differentiation (day 1) ARID1B was upregulated while ARID1A was strongly repressed to levels barely detectable (Fig. 7a). ARID1B remained the only active ARID1 subunit between days 1–5 (i.e., during the formation of the neuroectodermal cells; Fig. 7a). Finally, on day 7, ARID1B was abruptly downregulated and ARID1A was restored at high levels on day 9. This latter switch corresponded to the beginning of the differentiation of the neuroectodermal spheres into migratory CNCCs (Fig. 7a). Together, these data suggested that throughout the course of CNCC differentiation, multiple switches between ARID1A and ARID1B occur and that these two ARID1 subunits regulate specific developmental stages during CNCC formation.

Next, we profiled ARID1A protein levels during iPSC-to-CNCC differentiation in patient-derived lines. Time-course immunoblotting revealed that the temporary decommissioning of ARID1A during neuroectoderm specification (~days 1–7) failed to occur in both of the patient cell lines (Fig. 7b). In fact, in the patient lines, ARID1A protein levels were maintained throughout the course of iPSC differentiation (days 1–11) (Fig. 7b).

In addition to perpetual expression, there was an approximately fivefold increase in ARID1A expression in patient cell lines, relative to controls at day 5 (Patient 19: 4.9-fold; Patient 26:

5.1-fold; Fig. 7c), a time point which exhibited robust ARID1B expression in control cells. Together, these data suggested that the patient cells compensated for the partial loss of ARID1B by maintaining aberrantly high ARID1A levels throughout the differentiation process.

A recent study conducted on liver cells demonstrated that ARID1A- and ARID1B-containing BAF may have antagonistic functions in the transcriptional regulation of specific genes, with ARID1B acting prevalently as a repressor of enhancer elements, as opposed to the ARID1A, which mostly functions as an activator[44]. Hence, we hypothesized that the perpetual and robust ARID1A protein levels detected in patient-derived cells during iPSC-to-CNCC differentiation, and specifically during neuroectoderm specification, may result in the prolonged activity of pluripotency enhancers. Consistent with the recent studies which reported that the (ARID1A-containing) esBAF regulates pluripotency genes in iPSCs and ESCs[4–6], ARID1A ChIP-seq performed at the iPSC stage revealed that the 5540 pluripotency enhancers and promoters were bound by ARID1A in all the four lines (Fig. 7d; number of ARID1A peaks in iPSC ranged from 11,634 to 26,434 depending on the replicates). Conversely, the same experiment conducted on day 5 of differentiation revealed that while ARID1A binding was lost at the 5540 pluripotency elements in control cell lines, it was maintained in patient cell lines (Fig. 7d; the number of ARID1A peaks at day 5 in the patient lines ranged from 2405 in P19 to 17,080 in P26, indicating a potentially more widespread ARID1A activity in the patient line showing the strongest molecular impairment). In summary, these data indicated that the status of a set of ~5500 pluripotency enhancers and promoters were regulated by ARID1A at the iPSC stage and by ARID1B during exit from pluripotency and neuroectodermal lineage commitment. Importantly, *ARID1B* haploinsufficiency in patient cell lines triggered a compensatory mechanism that resulted in persistent binding of ARID1A at pluripotency enhancers throughout differentiation.

**The ARID1B-BAF complex exclusively incorporates SMARCA4 as an ATPase subunit.** Finally, we addressed the composition of the ARID1B-BAF complex at day 5 of iPSC-to-CNCC differentiation. To do so, we performed immunoprecipitation of endogenous ARID1B followed by mass-spectrometry (IP-MS). In control cells, ARID1B co-eluted with a total of 10 additional BAF subunits (hereafter ARID1B-BAF; Supplementary Fig. 5a). In mammals, BAF complexes incorporate two widely interchangeable and mutually exclusive ATPase subunits, SMARCA2, and SMARCA4. Remarkably, SMARCA4 was the only ATPase subunit identified as coeluting with ARID1B in our IP-MS, while zero peptides of SMARCA2 were detected (Supplementary Fig. 5a). This suggests that ARID1B-BAF selectively incorporates only SMARCA4 as a catalytic subunit during CNCCs formation.

We repeated this experiment in the patient lines to determine if perpetually active ARID1A replaced ARID1B in the ARID1B-BAF or if instead, it was part of a completely different BAF configuration. We thus performed ARID1A IP-MS at day 5 in patient lines. Notably, in the two patient lines ARID1A co-eluted with all the other subunits of the ARID1B-BAF (Supplementary Fig. 5b). In summary, the perpetual presence of ARID1A-BAF in patient cells correlated with a very specific (and previously uncharacterized) configuration of the BAF complex (ARID1B-BAF), which was active during exit from pluripotency and neuroectoderm specification.

Intriguingly, the transcription factor SALL4 also co-eluted with ARID1B at day 5, suggesting a possible interaction with the complex. Like ARID1B, SALL4 is also dispensable for the maintenance of the pluripotency networks, whereas it is essential

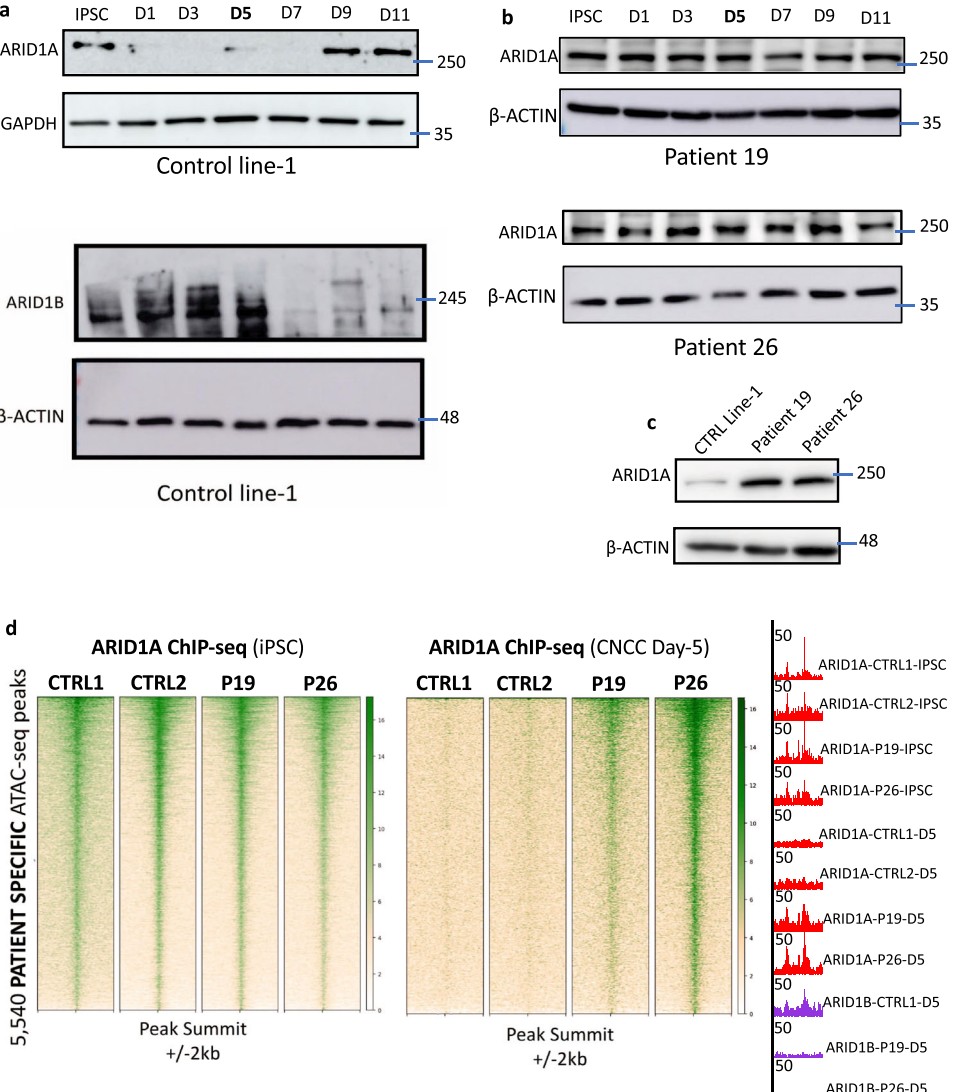

**Fig. 7 A switch between ARID1A-BAF and ARID1B-BAF is required for a successful exit from pluripotency. a** Time-course immunoblot conducted using Control Line-1 during CNCC differentiation shows that ARID1A is active at the iPSC stage, and abruptly downregulated at day 1 of differentiation. ARID1A protein level is upregulated again after day 7, mirroring ARID1B's downregulation at the same time point. Experiment repeated twice (different biological replicates). Marker bars display kDa. **b** ARID1A time-course immunoblot in the two patient lines. Experiment repeated twice (different biological replicates). Marker bars display kDa. **c** ARID1A immunoblot: both patient lines display aberrantly high ARID1A's protein level at CNCC Day 5. Experiment repeated twice (different biological replicates). Marker bars display kDa. **d** ChIP-seq for ARID1A in the two control lines and the two patient lines at iPSC stage (day 0) and at CNCC Day 5. Spike-in normalized heatmaps are centered on the 5540 pluripotency enhancers. The UCSC screenshot shows ARID1A and ARID1B binding at a patient-specific ATAC-seq peak (chr19:18,050,000–18,100,000) in the control and patient lines at iPSC stage and at CNCC Day 5.

for lineage commitment in early mammalian development, during which it targets sites with binding motifs also recognized by SOX2, OCT4, and NANOG[45–48]. SALL4 was previously shown to interact with the NuRD repressive complex[46], while interactions with BAF have been largely unexplored. It was recently demonstrated that this transcription factor has an affinity for AT-rich regions[49], thus providing further support to the ARID1B-SALL4 interaction. SALL4 mutations are also associated with developmental syndromes, including Okihiro syndrome, Holt-Oram syndrome, and Townes-Brocks Syndrome[50]. Notably, the SALL4 gene is downregulated in the Coffin-Siris patients at CNCC day 5 but not in undifferentiated iPSCs, suggesting a possible feedback mechanism between ARID1B and SALL4 during lineage commitment. Future studies will be necessary to support the speculation that SALL4 serves as an intermediator for ARID1B-BAF recruitment at the pluripotency enhancers.

## Discussion

ARID1B is a member of the evolutionarily conserved SWI/SNF (BAF) chromatin remodeler[22,51]. De novo haploinsufficient mutations in the *ARID1B* gene cause severe neurodevelopmental disorders, which affect both physical and cognitive development. In this study, we investigated the role of Coffin-Siris-associated *ARID1B* mutations in the context of craniofacial development and report the discovery of an unappreciated function of the BAF complex: attenuation of the gene expression program associated with pluripotency maintenance upon differentiation cues. We found that this repressive function is performed at pluripotency enhancers and promoters by a specific BAF complex configuration (ARID1B-BAF), which is composed of nine subunits, with the enzymatic activity seemingly carried out exclusively by SMARCA4. It is worth noting that future studies will be necessary to determine with enhancer repression is directly performed by

ARID1B, or if ARID1B's role is rather to prevent ARID1A from binding at these enhancers to keep them active

As a consequence of the *ARID1B* mutations, Coffin-Siris patient cells fail to repress the pluripotency elements. This subsequently elicits aberrant SOX2 activity genome-wide, which in turn leads to the upregulation of multiple pluripotency genes, including *NANOG* and its associated gene network, and to the downregulation of the genes responsible for coordinating the exit from pluripotency (PPARα/RXRα pathway). We demonstrated that these pluripotency enhancers are normally maintained in an active state by ARID1A-BAF at the iPSC stage, and subsequently repressed by the ARID1B-BAF throughout neuroectoderm formation, which is the first stage of CNCC development.

A switch between ARID1A-BAF and ARID1B-BAF upon differentiation cues is hence necessary for commitment towards the neuroectodermal lineage. We additionally report that a second switch from ARID1B-BAF to ARID1A-BAF takes place later, when the neuroectodermal spheres differentiate into CNCCs. This suggests that ARID1B is likely the dominant ARID1 subunit during neuroectoderm specification, whereas ARID1A regulates the following stage, during which the neuroectodermal spheres differentiate into migratory CNCCs. It is worth noting that the neuroectoderm not only gives rise to all regions of the brain and central nervous system (hindbrain, midbrain, forebrain, spinal cord, and motor neurons), but also gives rise to the neural crest cells, which emanate out from the dorsal aspect of the neural tube. Therefore, we speculate that the dysregulation of neuroectoderm specification caused by *ARID1B* mutations may underlie both the cognitive impairment and craniofacial abnormalities that are typical of Coffin-Siris syndrome. This model is further supported by a significant overlap of differentially expressed genes identified in our study (neural crest formation) and in a recently published *ARID1B*-KO-mouse model (brain tissue)[30].

Other studies have previously suggested that switches between SWI/SNF subunits play important roles in cell fate determination. For example, a switch between the two catalytic subunits SMARCA4 and SMARCA2 mediates the activation of human IFNγ-activated genes[52]. Similarly, a gain of the subunit BAF53a in the neuron-specific BAF (nBAF) is required to control cell cycle exit in developing neurons[2,3]. With our study, we discovered a binary switch between BAF subunits (ARID1A/ARID1B), critical for the exit from pluripotency. Importantly, a balance between pro-self-renewal and pro-differentiation signals is pivotal for the determination of stem cell fate[53]. We demonstrate that such balance is lost in Coffin-Siris patients, whose cells are unable to perform the ARID1A/ARID1B switch at the pluripotency enhancers at the onset of differentiation. This switch is essential to successfully complete CNCC differentiation.

Pluripotency is orchestrated by a transcription factor network that needs to be extinguished in an orderly manner to enable lineage commitment and differentiation[53–55]. We find that ARID1B-BAF plays an essential role in this process, by means of repressive activity at pluripotency enhancers for the SOX2, NANOG, and OCT4 networks. Similarly, an association between SOX3 and the SMARCA2 ATPase subunit of BAF was recently suggested in a study of neural development in the Nicolaides-Baraitser syndrome[56]. It is worth noting that Coffin-Siris and Nicolaides-Baraitser syndromes share many physical and neurological phenotypes[56–58].

The BAF complex is predominantly considered as a transcriptional activator, which balances the polycomb repressor complexes (PRC1, PRC2) in the modulation of gene expression[7,59]. Despite this widely held belief, repressive activity for BAF has been reported. For instance, a study conducted on hepatocellular carcinoma cell lines uncovered that ARID1A-containing BAF activates and represses roughly equal numbers of genes, while ARID1B-containing BAF was found to primarily repress enhancer activity[44]. Our experiments corroborate these findings, supporting an enhancer-repressor function for ARID1B-BAF. We demonstrate that the repressive activity of ARID1B-BAF is specific to a set of ~4900 enhancers and ~600 promoters, enriched for SOX2- and NANOG-binding sites. In control cells, these cis-regulatory elements are highly active at the iPSC stage, moderately active in the first four days of differentiation, and finally repressed by day 5, a time point at which we observed the peak of ARID1B protein expression. Coffin-Siris patient-derived iPSC lines exhibit aberrant chromatin accessibility at these cis-regulatory elements for many days after the onset of iPSC-to-CNCC differentiation, enforcing a perpetual pluripotency signature which persists even after 2 weeks of differentiation.

Patient 26-derived cells display the most extreme cellular and molecular phenotype, with a large population of cells remaining pluripotent at day 14 of differentiation, likely as a consequence of higher *SOX2, OCT4,* and *NANOG* expression and activity. Cells derived from this patient also show the highest levels of ARID1A binding at these enhancers at day 5 of CNCC differentiation. Although it is difficult to formally compare disease severity since there are no accepted severity scales for Coffin-Siris syndrome, it is worth noting that Patient 26 had a more severe disease process than Patient 19. For example, Patient 26 was not able to speak at 7 years, whilst Patient 19 started speaking at 4 years. In addition, Patient 26 was affected by pyloric stenosis, a congenital anomaly in the digestive tract thought to be associated with impaired migration of the enteric neural crest. We consider it unlikely that the difference is caused solely by the mutations in *ARID1B*, since both patients show a comparable reduction in ARID1B protein levels. We speculate that additional genetic factors may cooperate with *ARID1B* haploinsufficiency to determine the clinical severity of the syndrome. Even if our study is based on only two patients vs two control lines, multiple assays (ATAC-seq, ChIP-seq, RNA-seq, and differentiation assays) highlighted important differences during the differentiation of wild-type vs ARID1B-mutant iPSCs to CNCCs. However, additional experiments with a larger set of patient-derived cell lines would be required to support this model.

Furthermore, it would also be important to investigate other differentiation lineages to elucidate whether the ARID1A/ARID1B switch is only important for neural crest differentiation or if, instead, it represents a more widespread mechanism utilized by stem cells to exit the pluripotent state and undergo lineage commitment.

Finally, further investigations will be necessary to elucidate the mechanism(s) responsible for the repressive activity of ARID1B-BAF. Recent studies have demonstrated that the function of BAF (including ARID1A-BAF) as a transcriptional activator is mediated by the AP-1 transcription factors[56,60,61]. On the other hand, little is known of potential co-factors mediating the repressive function of ARID1B-BAF.

## Methods

**Antibodies**. ARID1B ChIP-Seq: Abcam ab57461. ARID1B western blot: Santa-Cruz sc-32762 and Abcam ab57461. ARID1A ChIP-Seq: GeneTex GTX129433. ARID1A western blot: Cell Signaling Technologies 12354 S. Beta-Actin western blot: Cell Signaling Technologies 8457 P. SOX2 ChIP-Seq: Active Motif 39843. NANOG ChIP-Seq: R&D Systems AF1997. H3K27ac ChIP-Seq: Abcam ab4729. GAPDH western blot: Cell Signaling Technologies 5174 T. CD10 Flow Cytometry: Miltenyi Biotech 130-124-262. CD99 Flow Cytometry: Miltenyi Biotech 130-121-086. SSEA4 Flow Cytometry: Biolegend 330417. TRA-1-60-R Flow Cytometry: Biolegend 330609. IgG ChIP-qPCR: Cell Signaling Technologies 2729 S. Cell Signaling HRP-conjugated anti-rabbit (7074 S) and anti-mouse (7076 S) were used as secondary antibodies in western blot. Spike-in Antibody: Active Motif 61686. Spike-in Chromatin: Active Motif 53083. Antibodies used in immunofluorescence: Anti-Mouse OCT4 (STEMCELL TECHNOLOGIES, 60059, 1:200); Rabbit Monoclonal Anti-Sox9 (abcam, ab185230,1:250); pPolyclonal Goat Anti-Nanog (R&D System, AF1997-SP,1:20); donkey anti-goat IgG (H + L) Alexa 488 (Jackson ImmunoResearch, 705-545-003, 1:500); donkey anti-mouse IgG (H + L) Alexa 647 (Jackson ImmunoResearch, 715-605-150, 1:500); Donkey anti-rabbit IgG (H + L) Cy3 (Jackson ImmunoResearch, 11-165-152, 1:500)., Monoclonal mouse anti-

human TRA-1-60 Antibody (Millipore MAB4360C3,1:100), Monoclonal mouse Anti-Stage-Specific Embryonic Antigen-4 Antibody (Millipore, MAB4304,1:100), Polyclonal Goat Anti- Human/Mouse Oct-3/4 Antibody (R&D System, AF1759,1:20).

**Human iPSC culture**. Control iPSC lines were obtained from the iPSC Core of the University of Pennsylvania (Control line-1: SV20 line, male, age 43) and from the Coriell Institute for Medical Research (Camden, NJ. Control line-2: GM23716, female, age 16). Skin fibroblasts from the two pediatric Coffin-Siris patients (one teenager one young adult) were obtained by the team of Dr. Gijs Santen at Leiden University. Patient 19 is a female, while Patient 26 is a male.

For the usage of the patient samples, we complied with all relevant ethical regulations for work with human participants, and that informed consent was obtained by Leiden University (LUMC), which approved the protocol under the coordination of Dr. Gijs Santen. The study was conducted in accordance with the criteria set by the Declaration of Helsinki. The protocol to make iPSCs was approved by the IRB at LUMC. Collecting patient material and establishing iPSCs have all been performed according to local (LUMC) IRB protocols.

The fibroblasts were reprogrammed into iPSCs with the polycistronic lentiviral vector LV.RRL.PPT.SF.hOKSM.idTomato.-preFRT by LUMC human iPSC Hotel as described elsewhere[62,63].

Multiple clones per line were derived. For each clone, pluripotency was assessed by immunofluorescence microscopy using antibodies against NANOG, OCT3/4, SSEA4, and Tra-1-81 under maintenance conditions and antibodies against (TUBB3, AFP, and CD31) after spontaneous differentiation into the three germ layers as described elsewhere[62]. Clones with proper pluripotent characteristics were selected for downstream usage. Karyotyping by G binding was assessed for all the four lines by the Leiden University Medical Center and short tandem repeat (STR) profiling was performed by the Leiden University Medical Center and then replicated by the Stem Cell and Regenerative Neuroscience Center at Thomas Jefferson University. The iPSC lines were expanded in feeder-free, serum-free mTeSR™1 medium (STEMCELL Technologies). Cells were passaged ~1:10 at 80% confluency using ReLeSR (STEMCELL Technologies) and small cell clusters (50–200 cells) were subsequently plated on tissue culture dishes coated overnight with Geltrex™ LDEV-Free hESC-qualified Reduced Growth Factor Basement Membrane Matrix (Fisher-Scientific).

**CNCC differentiation**. The iPSC lines were differentiated into CNCC as previously described[24]. In brief, iPSCs were treated with CNCC Derivation media: 1:1 Neurobasal medium/D-MEM F-12 medium (Invitrogen), 0.5× B-27 supplement with Vitamin A (50× stock, Invitrogen), 0.5× N-2 supplement (100× stock, Invitrogen), 20 ng/ml bFGF (Biolegend), 20 ng/ml EGF (Sigma-Aldrich), 5 µg/ml bovine insulin (Sigma-Aldrich) and 1× Glutamax-I supplement (100× stock, Invitrogen). Medium (3 ml) was changed every day. Three days after the appearance of the migratory CNCC, cells were detached using accutase and placed into geltrex-coated plates. The early migratory CNCCs were then transitioned to CNCC early maintenance media: 1:1 Neurobasal medium/D-MEM F-12 medium (Invitrogen), 0.5× B-27 supplement with Vitamin A (50× stock, Invitrogen), 0.5× N-2 supplement (100× stock, Invitrogen), 20 ng/ml bFGF (Biolegend), 20 ng/ml EGF (Sigma-Aldrich), 1 mg/ml bovine serum albumin, serum replacement grade (Gemini Bio-Products # 700-104 P) and 1× Glutamax-I supplement (100× stock, Invitrogen).

**ARID1B Knockdown**. To make concentrated lentivirus, HEK293T cells were transfected with a pLenti plasmid in which we cloned an shRNA for *ARID1B* (GPP Web Portal: TRCN0000107361). iPSCs were lentivirally transduced by incubating the cells with concentrated virus overnight at 37° C. The next morning the media was changed, and 2 mg/ml puromycin (InvivoGen) were added 24 h after infection. After 72 h, the iPSCs that survived the selection were then differentiated in CNCC using the above-described protocol, and collected at Day 5 for the genomic experiments. The cells were kept under puromycin selection for the entire duration of the differentiation. The knock-down efficiency was quantified via western blot.

**Flow cytometry analysis of surface markers**. To obtain a single-cell suspension for flow cytometry analysis, control and patient cells were treated with Accutase for 5 min. Cells were then washed with cold phosphate-buffered saline (PBS)-2% fetal bovine serum (FBS) and live cells were counted. $1 \times 10^6$ cells/condition were resuspended in 100 µL PBS-2% FBS and stained. For pluripotency evaluation, 4 µl of the respective antibodies were used: APC anti-human SSEA4 antibody (Biolegend, #330417) and PE anti-human TRA-1-60-R antibody (Biolegend, #330609). For analysis of differentiation, 2 µl of the respective antibodies were used: FITC anti-human CD10 (Miltenyi Biotec, #130-124-262) and APC anti-human CD99 (Miltenyi Biotec, #130-121-096). Cells were incubated for 15 min on ice and protected from light, before transferring them into FACS tubes containing an additional 300 µL PBS-2% FBS. Flow cytometry data were acquired using a BD LSR II flow cytometer and analyzed with FlowJo Software version 10.7.

**Western blot**. For total lysate, cells were harvested and washed three times in 1× PBS and lysed in radioimmunoprecipitation assay buffer (RIPA buffer) (50 mM Tris-HCl pH7.5, 150 mM NaCl, 1% Igepal, 0.5% sodium deoxycholate, 0.1% SDS,

500 µM DTT) with proteases inhibitors. Twenty µg of whole cell lysate were loaded in Novex WedgeWell 4–20% Tris-Glycine Gel (Invitrogen) and separated through gel electrophoresis (SDS–PAGE) Tris-Glycine-SDS buffer (Invitrogen). The proteins were then transferred to ImmunBlot PVDF membranes (ThermoFisher) for antibody probing. Membranes were incubated with 10% BSA in TBST for 30 min at room temperature (RT), then incubated for variable times with the suitable antibodies diluted in 5% BSA in 1× TBST, washed with TBST, and incubated with a dilution of 1:10000 of secondary antibody for one hour at RT. The antibody was then visualized using Super Signal West Dura Extended Duration Substrat (ThermoFisher) and imaged with Amersham Imager 680. Full membranes (uncropped) of the blots presented in the main figures are available as source data file.

**Cell fractionation**. In all, $5 \times 10^6$ cells/condition were collected and suspended in E1 buffer (50 mM HEPES-KOH, 140 mM NaCl, 1 mM EDTA, 10% glycerol, 0.5% NP-40, 0.25% Triton X-100, 1 mM DTT, 1× Proteinase Inhibitor) followed by a centrifugation step of $1100 \times g$ at 4 °C for 2 min. The cytoplasmic fraction was collected in a fresh tube. Cells were washed two more times with E1 buffer. Pellet was subsequently suspended in E2 buffer (10 mM Tris-HCl, 200 mM NaCl, 1 mM EDTA, 0.5 mM EGTA, 1× Proteinase Inhibitor) followed by a centrifugation step of $1100 \times g$ at 4 °C for 2 min. The nuclear fraction was collected in a fresh tube. Cells were washed two more times with E2 buffer. After the third wash, the pellet was suspended in E3 buffer (500 mM Tris-HCl, 500 mM NaCl, 1× Proteinase Inhibitor) and sonicated for 15 sec (5 sec ON/ 5 sec OFF). Cytoplasmic, nuclear, and chromatin fractions were centrifuged at $16,000 \times g$ for 10 min at 4 °C.

**Immunofluorescence**. Upon fixation (4% PFA for 10 min), cells were permeabilized in blocking solution (0.1% Triton X-100, re PBS, 5% normal donkey serum) and then incubated with the antibody of interest. The total number of cells in each field was determined by counterstaining cell nuclei with 4,6-diamidine-2-phenylindole dihydrochloride (Sigma-Aldrich; 50 mg/ml in PBS for 15 min at RT). Immunostained cells were mounted in Aqua-Poly/Mount (Polysciences) and analyzed at epi-fluorescent or confocal microscopy, using a Nikon A1R+. Images were captured with a ×40 objectives and a pinhole of 1.2 Airy unit. Analyses were performed in sequential scanning mode to rule out cross-breeding between channels.

**Real-time quantitative polymerase chain reaction (RT-qPCR)**. Cells were lysed in Tri-reagent and RNA was extracted using the Direct-zol RNA MiniPrep kit (Zymo Research). In all, 600 ng of template RNA was retrotranscribed into cDNA using RevertAid first-strand cDNA synthesis kit (Thermo Scientific) according to manufacturer directions. 15 ng of cDNA were used for each real-time quantitative PCR reaction with 0.1 µM of each primer, 10 µL of PowerUp™ SYBR™ Green Master Mix (Applied Biosystems) in a final volume of 20 µl, using QuantStudio 3 Real-Time PCR System (Applied Biosystem). Thermal cycling parameters were set as follows: 3 min at 95 °C, followed by 40 cycles of 10 s at 95 °C, 20 s at 63 °C followed by 30 s at 72 °C. Each sample was run in triplicate. 18 S rRNA was used as normalizer. Primer sequences are reported in Supplementary Data 7.

**ChIP-Seq and ChiP-qPCR**. Samples from different conditions were processed together to prevent batch effects.

For SOX2, NANOG, and H3K27ac, for each replicate, 10 million cells were cross-linked with 1% formaldehyde for 5 min at RT, quenched with 125 mM glycine, harvested, and washed twice with 1× PBS. The pellet was resuspended in ChIP lysis buffer (150 mM NaCl, 1% Triton X-100, 0,7% SDS, 500 µM DTT, 10 mM Tris-HCl, 5 mM EDTA) and chromatin was sheared to an average length of 200–500 bp, using a Covaris S220 Ultrasonicator. The chromatin lysate was diluted with SDS-free ChIP lysis buffer. For ChIP-seq, 10 µg of antibody (3 µg for H3K27ac) was added to 5 µg of sonicated chromatin along with Dynabeads Protein A magnetic beads (Invitrogen) and incubated at 4 °C overnight. For SOX2 and NANOG ChIP-seq, 10 ng of spike-in Drosophila chromatin (Active Motif) was added to each sample with 2 µg spike-in antibody (Active Motif). On day 2, beads were washed twice with each of the following buffers: Mixed Micelle Buffer (150 mM NaCl, 1% Triton X-100, 0.2% SDS, 20 mM Tris-HCl, 5 mM EDTA, 65% sucrose), Buffer 500 (500 mM NaCl, 1% Triton X-100, 0.1% Na deoxycholate, 25 mM HEPES, 10 mM Tris-HCl, 1 mM EDTA), LiCl/detergent wash (250 mM LiCl, 0.5% Na deoxycholate, 0.5% NP-40, 10 mM Tris-HCl, 1 mM EDTA) and a final wash was performed with 1× TE. Finally, beads were resuspended in 1× TE containing 1% SDS and incubated at 65 °C for 10 min to elute immunocomplexes. Elution was repeated twice, and the samples were further incubated overnight at 65 °C to reverse cross-linking, along with the untreated input (5% of the starting material). On day 3, after treatment with 0.5 mg/ml Proteinase K for 1 h at 65 °C, DNA was purified with Zymo ChIP DNA Clear Concentrator kit and quantified with QUBIT.

For ARID1A and ARID1B ChIP-Seq, 10 million cells were cross-linked with EGS (150 mM) for 30 min at RT followed by a second cross-link with 1% formaldehyde for 15 min at RT. The formaldehyde was quenched by adding glycine (0.125 M) for 10 min at RT. Cells were washed twice with 1× PBS. Pellet was resuspended in buffer LB1 (50 mM Hepes-KOH, 140 mM NaCl, 1 mM EDTA, 10%

Glycerol, 0.5% NP-40, 0.255 Triton X-100), incubated 10 min at 4 °C followed by a centrifugation step of 600 g for 5 min at 4 °C. Pellet was suspended in buffer LB2 (10 mM Tris-HCl, 20 mM NaCl, 1 mM EDTA, 0.5 mM EGTA) incubated 10 min at 4 °C followed by a centrifugation step of 600 g for 5 min at 4 °C. Cells were then resuspended in buffer LB3 (10 mM Tris-HCl, 200 mM NaCl, 1 mM EDTA, 0.5 mM EGTA, 0.1% Na-DOC, 0.5% N-laurosylsarcosine) incubated 10 min at 4 °C followed by a centrifugation step of 600 g for 5 min at 4 °C. Pellet was suspended in LB3 and chromatin was sheared to an average length of 200–500 bp, using a Covaris S220 Ultrasonicator. For each sample, 15μg of sonicated chromatin was incubated at 4 °C overnight along with Dynabeads Protein G conjugated with 10μg of antibody. On day 2, beads were washed once with each of the following buffers: WB1 (50 mM Tris-HCl, 150 mM NaCl, 0.15 SDS, 0.1% Na-DOC, 1% Triton X-100, 1 mM EDTA), WB2 (50 mM Tris-HCl, 500 mM NaCl, 0.15 SDS, 0.1% Na-DOC, 1% Triton X-100, 1 mM EDTA), WB3 (10 mM Tris-HCl, 250 mM LiCL, 0.55 NP-40. 0.55 Na-DOC, 1 mM EDTA), TE Buffer (10 mM Tris-HCl, 1 mM EDTA). Finally, beads were resuspended in EB (10 mM tris-HCl, 0.55 SDS, 300 mM NaCl, 5 mM EDTA) and incubated at 65 °C for 30 min to elute immunocomplexes. Elution was repeated twice, and the samples were further incubated overnight at 65 °C to reverse cross-linking, along with the untreated input (5% of the starting material). On day 3, after treatment with 0.5 mg/ml Proteinase K for 1 h at 65 °C.

For all ChIP-seq experiments, barcoded libraries were made with NEB ULTRA II DNA Library Prep Kit for Illumina, and sequenced on Illumina NextSeq 500, producing 75 bp SE reads.

For ChIP-qPCR, on day 1 the sonicated lysate was aliquot into single immunoprecipitations of $2.5 \times 10^6$ cells each. A specific antibody or a total rabbit IgG control was added to the lysate along with Protein A magnetic beads (Invitrogen) and incubated at 4 °C overnight. On day3, ChIP eluates and input were assayed by real-time quantitative PCR in a 20 μl reaction with the following: 0.4 μM of each primer, 10 μl of PowerUp SYBR Green (Applied Biosystems), and 5 μl of template DNA (corresponding to 1/40 of the elution material) using the fast program on QuantStudio qPCR machine (Applied Biosystems). Thermal cycling parameters were: 20 sec at 95 °C, followed by 40 cycles of 1 sec at 95 °C, 20 sec at 60 °C.

**ChIP-seq analyses**. After removing the adapters, the sequences were aligned to the reference hg19, using Burrows-Wheeler Alignment tool, with the MEM algorithm[64]. Aligned reads were filtered based on mapping quality (MAPQ > 10) to restrict our analysis to higher quality and likely uniquely mapped reads, and PCR duplicates were removed. We called peaks for each individual using MACS2[65] (H3K27ac) or Homer[66], at 5% FDR, with default parameters.

**RNA-seq**. Cells were lysed in Tri-reagent (Zymo research) and total RNA was extracted using Quick-RNA Miniprep kit (Zymo research) according to the manufacturer's instructions. RNA was further quantified using DeNovix DS-11 Spectrophotometer while the RNA integrity was checked on Bioanalyzer 2100 (Agilent). Only samples with RIN value above 8.0 were used for transcriptome analysis. RNA libraries were prepared using 1 μg of total RNA input using NEB-Next® Poly(A) mRNA Magnetic Isolation Module, NEBNext® UltraTM II Directional RNA Library Prep Kit for Illumina® and NEBNext® UltraTM II DNA Library Prep Kit for Illumina® according to the manufacturer's instructions (New England Biolabs).

**RNA-Seq analyses**. Reads were aligned to hg19 using STAR v2.5[67], in 2-pass mode with the following parameters: --quantMode TranscriptomeSAM --out-FilterMultimapNmax 10 - -outFilterMismatchNmax 10 --out-FilterMismatchNoverLmax 0.3 --alignIntronMin 21 -- alignIntronMax 0 --alignMatesGapMax 0 --alignSJoverhangMin 5 --runThreadN 12 -- twopassMode Basic --twopass1readsN 60000000 --sjdbOverhang 100. We filtered bam files based on alignment quality ($q$ = 10) using Samtools v0.1.19[64]. We used the latest annotations obtained from Ensembl to build reference indexes for the STAR alignment. Kallisto[68] was used to count reads mapping to each gene. RSEM[69] was instead used to obtain FPKM (Fragments Per Kilobase of exon per Million fragments mapped). We analyzed differential gene expression levels with DESeq2[70], with the following model: design = ~condition, where condition indicates either CTRL or Patients.

**ATAC-Seq**. For ATAC-Seq experiments, 50,000 cells per condition were processed as described in the original ATAC-seq protocol paper[71]. ATAC-seq data were processed with the same pipeline described for ChIP-seq, with one modification: all mapped reads were offset by +4 bp for the forward-strand and −5 bp for the reverse-strand. After peak calling (MACS2), peaks replicated in all four lines (hereafter consensus peaks) were used for downstream analyses.

**Nuclear extract, IP, and LC-MS/MS**. After collection, cells were washed twice with ice-cold PBS before resuspension in co-IP buffer (20 mM Tris pH 7.9, 100 mM NaCl, 0.1% NP-40, 0.5 mM dithiothreitol (DTT), protease inhibitors), and rotated for 5 min at 4 °C. After spinning down at $380 \times g$ for 10 min, the nuclear pellet was resuspended in buffer C (20 mM Tris pH 8.0, 1.5 mM MgCl₂, 0.42 M NaCl, 25% glycerol, 0.2 mM EDTA, 0.5 mM DTT, protease inhibitors), dounce homogenized (with B pestle), and incubated at 4 °C for 30 min. The extract was centrifuged at

$13{,}000 \times g$ for 30 min, and the supernatant was kept as a nuclear extract. The nuclear extract was dialyzed overnight in BC80 (20 mM Tris pH 8.0, 80 mM KCl, 0.2 mM EDTA, 10% glycerol, 1 mM B-mercaptoethanol, 0.2 mM phenylmethylsulfonyl fluoride), cleared, and stored at −80 °C. For the IP, 1.5 mg of nuclear extract was incubated for 3 h at 4 °C with 6 μg ARID1B antibody and 50 μL of Dynabeads Protein A, and the control IP was performed with 0.75 mg of nuclear extract and 25 μL of Dynabeads Protein A. Beads were washed three times with co-IP buffer, followed by a final wash with 0.05% NP-40 in PBS. Elution was performed by agitation in 0.1 M glycine pH 3.0 for one minute, and 1 M Tris base pH 11.0 was added to neutralize the pH of the eluate. Eluates were prepared for SDS–PAGE and run on a Novex WedgeWell 10% Tris-Glycine Gel (Invitrogen) with Tris-Glycine-SDS buffer (Bio-Rad), at 110 V for 10 min. The gel was stained with Colloidal Blue staining kit (Invitrogen), and further processed at the proteomics facility at the Wistar Institute. In brief, the gel lanes were excised, reduced with TCEP, alkylated with iodoacetamide, and digested with trypsin. Tryptic digests were analyzed using LC-MS/MS (a standard 90 min LC gradient on the Thermo Q Exactive HF mass spectrometer). MS/MS spectra were searched with full tryptic specificity against the UniProt human database (10/02/2020) using MaxQuant 1.6.17.0, and also searched for the common protein N-terminal acetylation, Asn deamidation, and Met oxidation. The protein and peptide false discovery rate was set at 1%.

**Statistical and genomic analyses**. All statistical analyses were performed using R v3.3.1. BEDtools v2.27.1[72] was used for genomic analyses. Pathway analysis was performed with Ingenuity-Pathway Analysis Suite (QIAGEN Inc., https://www.qiagenbioinformatics.com/products/ingenuity-pathway-analysis). Motif analyses were performed using the Meme-Suite[73], and specifically with the Meme-ChIP application. Fasta files of the regions of interest were produced using BED-Tools v2.27.1. Shuffled input sequences were used as background. $E$ values <0.001 were used as a threshold for significance[73].

**Reporting summary**. Further information on research design is available in the Nature Research Reporting Summary linked to this article.

## Data availability
The data that support this study are available from the corresponding author upon reasonable request. The original genome-wide data generated in this study have been deposited in the GEO database under accession code GSE169654. The proteomic data generated in this study have been deposited in ProteomeXchange Consortium via the PRIDE partner repository under the accession code PXD028557. Source data are provided with this paper.

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

## Acknowledgements

This Open Access publication was made possible in part by support from the Thomas Jefferson University Open Access Fund. The authors are deeply grateful to the Coffin-Siris patients and their families, who donated the samples for this research. The authors thank the physicians and technicians who collected and cultured the skin samples. This work was funded by the G. Harold and Leila Y. Mathers Foundation (M.T.), and the Gisela Thier Fellowship (G.W.E.S.). B.C. and P.P. were supported by NIH grants R01-HL127895 and R01 CA257251. The authors thank the Genomic Facilities at Thomas Jefferson University and at the Wistar Institute for the sequencing effort. The authors are grateful to Kelly Vonk (Leiden University) for the help with experimental procedures in the iPSC generation. An invaluable support was provided to the authors by the Stem Cell and Regenerative Neuroscience Center at Thomas Jefferson University, and in particular by Dr. Elizabeth Kropf. At last, M.T. and G.W.E.S. are thankful to Dr. Samantha Vergano (Children's Hospital of the King Daughters, Virginia, US) for making this collaboration and this work possible.

## Author contributions

M.T., G.W.E.S. and L.P. designed the project. G.W.E.S. recruited the patients and obtained the skin fibroblasts. H.M.M.M. and C.F. reprogrammed the patient fibroblasts into iPSCs and assessed their quality. L.D. performed initial iPSC characterization experiments. L.P. performed most of the experiments, with crucial contributions from P.P. A.T.C., C.S., C.A.O., S.O. and S.A.W. contributed to specific experiments (flow cytometry, immunoblots, mass-spectrometry). B.C. provided intellectual contribution and financial support to P.P. S.A.B. provided helpful discussion and editing. M.T., L.P., P.P. and S.D. analyzed the data. M.T., G.W.E.S., P.P. and L.P. wrote the manuscript, which was read and approved by all the authors.

## Competing interests

The authors declare no competing interests.
