## [Peer Review File · Nature Communications]

REVIEWER COMMENTS

Reviewer #1 (Remarks to the Author):

Pagliaroli et al., have submitted a manuscript detailing a developmental switch in BAF complexes as iPSC cells differentiate towards the CNCC lineage. ARID1B appears to be responsible for this switch as iPSCs from patients with ARID1B heterozygous mutation demonstrate deficiencies in CNCC differentiation that result from failed repression of pluripotency enhancers. The authors utilize a variety of genomic methodology to demonstrate that in ARID1B mutant cells, these enhancers maintain activating ARID1A-BAF complexes at these enhancers, exhibit features of open chromatin, and maintain expression of pluripotent genes. This is a very interesting study and would be of interest to wide fields of chromatin, epigenetic, and developmental biology researchers. I believe the study will need more controls and analysis within the experimental methodology to explain the conclusions described in the manuscript. My questions and concerns are listed below.

- 1) The western blots showing ARID1B increase during the CNCC differentiation time-course (Figure 2C) need some quantitation and more than one replicate. The Figure 2C blot looks like it is reused in Figure 6A. What is the quantitation of ARID1B protein in the patient chromatin fraction relative to control (Figure S1C) and is there any truncated protein product in the patient chromatin or total nuclear fraction? Why is there a doublet for ARID1B?
- 2) ARID1B is highly expressed in control iPSCs (Figure 2C). Does ARID1B associate with the patient specific ATAC-seq peak loci (pluripotent enhancers) in control iPSCs and how does the ARID1B binding change from iPSC to Day 5 CNCC?
- 3) The H3K27ac loss from iPSC to day 5 CNCC appears to occur in a small subset of enhancers while the majority of patient specific ATAC sites have low level H3K27ac that might not change across differentiation (Figure 3F). Although there is a trend of increased H3K27ac in the patient lines (Figure 3G), how many of the patient specific ATAC-peaks have significantly elevated patient H3K27ac read counts over control based on DESeq at day 5 of CNCC differentiation?
- 4) The expression data (Figure 4A) is confusing as the patient cells largely exhibit a gain of ATAC open chromatin (Figure 3A) but the majority of misregulated genes are losing expression rather than gaining expression (Figure 4A). Why is that and what percentage of the patient specific ATAC-seq peaks demonstrate an increase in nearby gene expression in patient cells? Are these important pluripotency genes?
- 5) In Figure 6C, what are these regions and why were they chosen? ARID1A ChIP-seq was performed on day 5 CNCC differentiated patient cells (Figure 6D). It should also be performed on control iPSC cells and control day 5 CNCC for comparison. Is the patient day 5 CNCC ARID1A elevation (Figure 6B) also observed in the iPSC lines prior to differentiation?
- 6) Some of the flow cytometry staining does not look to be impacted by CNCC differentiation. For example, many of the control day 14 CNCC cells have similar SSEA-4 levels as control iPSC cells (Figure 2D) and control day 14 CNCC CD10 levels are similar to control iPSCs (Figure 2E). Is this correct and why? The ARID1B heterozygous patient P19 and P26 cell lines have largely lost pluripotency (Figure 2D: 95% and 80% respectively) and differentiated (Figure 2E) by day14. Do the ARID1B +/- day 5 chromatin phenotypes throughout the manuscript represent a delay in pluripotency enhancer repression that ultimately gets resolved by day14? At day 14, are the majority of P19/P26 cells OCT4/Nanog negative and TFAP2A/NR2F1/SOX9 positive?
- 7) SALL4 is strongly hypothesized by the authors to mediate the ARID1B repressive mechanism. There is not experimental evidence in the manuscript to support this. It should be omitted from the manuscript unless there is more supporting evidence.

8) How does the author's model (Figure 7) fit in with published literature in BAF mutant animal models related to neural crest function?

Reviewer #2 (Remarks to the Author):

In this current manuscript, Trizzino et al reprogrammed Coffin-Siris patient-derived skin fibroblasts into iPSCs and used it as a model system to study the function of ARID1B in directing cranial neural crest cell (CNCC) differentiation. They found that patient derived iPSCs showed impaired differentiation compared to control iPSC cell lines. Mechanistically, they found that due to the haploinsufficiency of ARID1B, cells are unable to switch from ARID1A-BAF to ARID1B-BAF during induced differentiation, thus ARID1A-BAF remains at enhancers that regulating genes essential for stemness/pluripotency. Overall, the authors presented some interesting phenotypic discovery, while the mechanistic component lacks multiple critical controls. The enthusiasm is further reduced due to insufficient data analysis, and over-statement of some of the claims such as 'novel' ARID1B-BAF composition. If a key point of the model (also indicated in the title) is to demonstrate the switch of ARID1A to ARID1B-BAF complex during differentiation, it would be essential to have ChIP-seq of ARID1A, ARID1B in iPSCs and differentiated cells (Day 5) both in control and patient cell lines, as well as ATAC-seq (and K27Ac, if the authors want to add the enhancer component story). Completion of these experiments will allow systematic cross comparison of ARID1A, ARID1B at stem cell states and CNCC states, correlate their targeting to chromatin accessibilities (and enhancers), as well as SOX2 and NANOG targeting. If the model stands, we would expect loss of ARID1A targeting (or altered), gained ARID1B targeting, correlations with changes on chromatin accessibilities, and how do they collectively regulate gene expression programs. The current data is not sufficient to support the model. Importantly, throughout the manuscript, critical control experiments are missing in multiple places: In Fig1. Should show similar staining on control iPSCs so that readers can compare the patient derived iPSCs (Fig1d, Fig1) to. In Fig3. Why align ARID1B peaks called in control lines to patient specific (gained) ATAC-seq peaks, when there is low ARID1B in patient-derived cell lines? Several controls are missing: ATAC-seq at iPSCs stage both in control and patient cells; ARID1B ChIP-seq in control iPSCs at Day 0 and Day 5/7. Ideally, one would compare ATAC-seq changes of Day 5 vs Day 0 in control cell line and patient cell lines; as well as compare control and patient cells at Day 0 and Day 5. In this case, the authors only compared Day 5 peaks between control and patient cell line. Similarly, while it's nice the authors performed time-course K27Ac ChIP-seq in the control cell line, it is critical to have that data in the patient cell lines. Do these regions maintain active during differentiation? Are there any de novo enhancers in the patient samples? In Fig.4. Similarly, important controls are missing: RNA-seq of control and patient cell lines at Day 0, which will allow comparison/analysis of gene expression changes upon differentiation in the patient samples. In Fig6B: A time-course WB of ARID1A level in patient cells would be essential to assess how ARID1A protein level changes in the context of ARID1B haploinsufficiency. In Fig6E: it is an over-statement of identifying a 'novel' ARID1B-BAF complex composition. Looking at the listed subunits, all of them are known BAF complex members, while SMARCA5 is NOT a SWI/SNF subunit. While the authors claim only SMARCA4 is found in ARID1B IP, did they check the expression level of SMARCA2? Another control would be ARID1A mass-spec.

Minor points:

Fig6A is a duplicate of Fig2c.

FigS2A: What is the value of Y-axis? Why is it drastically different from FigS2C?

FigS2B: Why would control shRNA have an effect on knocking down ARID1B expression?

FigS3: Should also show the level of ARID1B and adding IgG control

Reviewer #3 (Remarks to the Author):

Pagliaroli et al. show that iPSCs from Coffin-Siris patients with heterozygous ARID1B mutations fail to undergo an ARID1A-to-ARID1B switch at the onset of induced cranial neural crest differentiation, resulting in impaired lineage determination. The authors conclude that impaired cranial neural crest differentiation underlies Coffin-Siris pathogenesis. BAF complex switching during differentiation and SWI/SNF complex roles in both maintenance of and exit from pluripotency are known. The strengths of the manuscript lie in the use of patient-derived cell lines and robust genomic experiments. Although the studies are well-done and manuscript is well-written, the physiological relevance of the findings are still unclear, especially since all of the studies were performed in iPSCs and little effort is made to connect the findings back to the disease pathogenesis.

-Please increase use of statistical tests in the figures.

-Please include descriptions of the statistical test used in the figure legends.

-With only an N=2 of patients, any associations with disease severity and differentiation defect are too speculative.

-Previous studies have shown Coffin-Siris-like defects in ARID1B haploinsufficient mice. In addition, it's unclear if mouse models or Coffin-Siris patients have defects in early central nervous system or neural crest development, or both. Based on the major conclusion raised in manuscript-that cranial neural crest differentiation underlies Coffin-Siris pathogenesis-it seems important to address the cellular etiology of the disease or at least determine if cranial neural crest differentiation genes identified are affected in vivo. This would help determine if the disease is associated with cranial neural crest defects and thus further support the genomic mechanisms put forth in the manuscript.

Reviewer#1

Pagliaroli et al., have submitted a manuscript detailing a developmental switch in BAF complexes as iPSC cells differentiate towards the CNCC lineage. ARID1B appears to be responsible for this switch as iPSCs from patients with ARID1B heterozygous mutation demonstrate deficiencies in CNCC differentiation that result from failed repression of pluripotency enhancers. The authors utilize a variety of genomic methodology to demonstrate that in ARID1B mutant cells, these enhancers maintain activating ARID1A-BAF complexes at these enhancers, exhibit features of open chromatin, and maintain expression of pluripotent genes. This is a very interesting study and would be of interest to wide fields of chromatin, epigenetic, and developmental biology researchers. I believe the study will need more controls and analysis within the experimental methodology to explain the conclusions described in the manuscript.

We thank the Reviewer for their kind words of appreciation of our study.

My questions and concerns are listed below.

1) The western blots showing ARID1B increase during the CNCC differentiation time-course (Figure 2C) need some quantitation and more than one replicate. The Figure 2C blot looks like it is reused in Figure 6A. What is the quantitation of ARID1B protein in the patient chromatin fraction relative to control (Figure S1C) and is there any truncated protein product in the patient chromatin or total nuclear fraction? Why is there a doublet for ARID1B?

As recommended by the Reviewer, we replicated the ARID1B time-course so that now Figures 2C and 7A (former 6A) show two different replicates of the blot. The two biological replicates are consistent in showing robust ARID1B protein level for the duration of the neuroectoderm formation (~first 5 to 7 days), after which ARID1B protein is decommissioned. There is some variability (expected across different biological batches of iPSC differentiation) regarding the day in which ARID1B peaks. We see the peak happening between day-5 and day-7, depending on the replicate. This perfectly correlates with the day in which the neuroectodermal spheres attach to the plate to start further differentiating into CNCCs. The day in which the neuroectodermal spheres attach varies between day-5 and day-7. Which is exactly when ARID1B peaks.

Also, as recommended by the Reviewer, we quantified using imageJ the ARID1B protein levels from the chromatin fraction (Figure S1a) to compare patients and control line. The quantifications are now reported in the manuscript, and show that, for both patients, over half of ARID1B protein is lost, compatible with haploinsufficiency.

There is no truncated protein product in any of the fractions (chromatin, nuclear, cytoplasm, see Figure S1a; Please not that the three fractions come from the same lysate fractionation), suggesting that the aberrant mRNA is degraded before translation, likely via non-sense mediated mRNA decay pathway, which is the most common biological pathway taking care of aberrant transcripts derived from haploinsufficient mutations.

To answer to the final question, there is a doublet for ARID1B because there are two annotated isoforms (~250 and 140 kDa). The two isoforms are equally affected by the patient mutations, since the aberrant STOP codon was generated before the alternative splicing site.

2) ARID1B is highly expressed in control iPSCs (Figure 2C). Does ARID1B associate with the patient specific ATAC-seq peak loci (pluripotent enhancers) in control IPSCs and how does the ARID1B binding change from iPSC to Day 5 CNCC?

As recommended by the Reviewer, we performed ChIP-seq for ARID1B at the iPSC stage and at the Day-5 in all the four lines (see Supplementary Figure S4b and Fig. 4d; the figures are also reported for convenience at the end of this paragraph).

According to these new experiments, there is no ARID1B binding at the pluripotency enhancers in any of the four lines at the iPSC stage (Supplementary Figure S4b). This was expected, because the BAF configurations reported as active in stem cells (esBAF, gBAF) do not include ARID1B as part of the complex.

At the day-5 (peak of ARID1B protein level) the picture is different. At this time point, the pluripotency enhancers are bound by ARID1B in the control lines, but not in the patient lines, which lack ARID1B (see new Fig. 4d).

3) The H3K27ac loss from iPSC to day 5 CNCC appears to occur in a small subset of enhancers while the majority of patient specific ATAC sites have low level H3K27ac that might not change across differentiation (Figure 3F). Although there is a trend of increased H3K27ac in the patient lines (Figure 3G), how many of the patient specific ATAC-peaks have significantly elevated patient H3K27ac read counts over control based on DESeq at day 5 of CNCC differentiation?

We thank the Reviewer for pointing this out. This reviewer's comment is in line with a comment we received from another Reviewer, who asked us to perform a time-course of H3K27ac after day-5 in the control and patient lines to better profile how these enhancers are differentially active between control and patient lines later during CNCC formation.

These experiments (data reported in Fig. 4g and also at the end of this paragraph for Reviewer's convenience) confirmed that at day-5 the differences in H3K27ac are still moderate between control and patient lines. In fact, at the day-5, like the Reviewer pointed out, the majority of the enhancers still display H3K27ac signal also in the control line, although such signal is twice as high in the two patient-lines compared to the control (see fig. below). And in fact, the DESEQ2 analysis only identified ~400 enhancers as significantly more active in patients.

On the other hand, at the day-7, the picture is markedly different. Namely, at this time-point these enhancers show NO signal in the control, while still displaying strong signal in both patient lines. Finally, at the day-9 these enhancers resulted inactivated (i.e. no H3K27ac signal) also in Patient-19, while they are still fully active in Patient-26. This is consistent with other experiments (flowcytometry, immunofluorescence, NANOG ChIP-seq) which showed a more "extreme" molecular phenotype for the patient-26. In some ways, the

patient-19 seems able to cope better than patient 26, but future studies, with a larger set of patients, will be needed to understand the patient-to-patient variability.

4) The expression data (Figure 4A) is confusing as the patient cells largely exhibit a gain of ATAC open chromatin (Figure 3A) but the majority of misregulated genes are losing expression rather than gaining expression (Figure 4A). Why is that and what percentage of the patient specific ATAC-seq peaks demonstrate an increase in nearby gene expression in patient cells? Are these important pluripotency genes?

Premise: Former Figure 4 is now Figure 5.

We apologize if these specific results were presented in a confusing manner. As correctly mentioned by the Reviewer, 71% of ALL the 2,365 differentially expressed genes were downregulated, while only 29% were upregulated. This could seem counter-intuitive, considering that ARID1B-haloin insufficiency opens up thousands of enhancers (patient-specific ATAC-seq peaks). However, if we focus exclusively on the in the subset of differentially expressed genes that are also the nearest gene to a patient-specific ATAC-seq peak (598/2,365 DE genes), then the picture is different: 55% are upregulated and 45% downregulated.

These 598 genes are important pluripotency genes, as shown by the pathway analysis (Figure 5B), that is based on these 598 genes and not on all the 2,365 DE genes.

We apologize if this latter point was not clear. We updated the figure and the manuscript accordingly, to make sure it is clearly stated that the pathway analysis was performed on the 598 DE genes that are also target of patient-specific ATAC-seq peaks.

5a) In Figure 6C, what are these regions and why were they chosen?

Former Figure 6 is now figure 7. The regions were four pluripotency enhancers that we randomly selected from the list of patient-specific ATAC-seq peaks. That said, we removed the ChIP-qPCR data from the paper since we now have ARID1A ChIP-seq data for all the four lines both in iPSCs and CNCC day-5 (Figure 7).

5b) ARID1A ChIP-seq was performed on day 5 CNCC differentiated patient cells (Figure 6D). It should also be performed on control iPSC cells and control day 5 CNCC for comparison. Is the patient day 5 CNCC ARID1A elevation (Figure 6B) also observed in the iPSC lines prior to differentiation?

As mentioned in point 5a, as requested by the Reviewer we have performed ARID1A ChIP-seq in the four lines at both iPSC stage and CNCC day-5 (Figure 7 and also reported at the end of this paragraph for Reviewer's convenience). These new data confirm that the 5,540 pluripotency enhancers and promoters are bound by ARID1A at the iPSC stage in all four

lines with no significant differences across lines. This is expected, because the esBAF (i.e. “embryonic stem cell BAF) incorporates exclusively ARID1A, and NOT ARID1B, as AT-rich subunit.

On the other hand, the ARID1A ChIP-seq at the day-5 confirmed that the two patient lines maintain aberrant ARID1A binding at these regions, while the control lines have completely lost ARID1A, replacing it with ARID1B (see Figures 7d and 4d).

6) Some of the flow cytometry staining does not look to be impacted by CNCC differentiation. For example, many of the control day 14 CNCC cells have similar SSEA-4 levels as control iPSC cells (Figure 2D) and control day 14 CNCC CD10 levels are similar to control iPSCs (Figure 2E). Is this correct and why? The ARID1B heterozygous patient P19 and P26 cell lines have largely lost pluripotency (Figure 2D: 95% and 80% respectively) and differentiated (Figure 2E) by day14. Do the ARID1B +/- day 5 chromatin phenotypes throughout the manuscript represent a delay in pluripotency enhancer repression that ultimately gets resolved by day14? At day 14, are the majority of P19/P26 cells OCT4/Nanog negative and TFAP2A/NR2F1/SOX9 positive?

Following the Reviewer’s suggestion, we aimed at quantifying to what extent are the patient CNCCs impaired/delayed in their differentiation. To this purpose, we performed Immunofluorescence (IF) for SOX9 (marker of neural crest identity), as well as OCT4 and NANOG (pluripotency markers). The IF was conducted on control line-1 and in the two patient lines at day-14 of iPSC-to-CNCC differentiation (i.e. endpoint of the protocol, when the CNCCs are supposed to be fully formed).

The IF (reported in the new Figure 3 and at the end of this paragraph for Reviewer’s convenience) revealed that nearly all the *ARID1B*^{+/-} patient cells are positive for the CNCC marker SOX9 but ALSO for NANOG. Further, one of the two patient-lines (P26) is also OCT4-positive in nearly all of the cells. This is in stark contrast with the control line, in which the SOX9⁺ cells are all NANOG-negative and OCT4-negative. This suggests that the patient CNCCs

have a pervasive and systemic issue with NANOG, that is aberrantly active from day-0 to day-14 in nearly all of the cells. This is consistent with the ATAC-seq (NANOG motif in the patient-specific ATAC-seq peaks), RNA-seq (NANOG-associated pathways as differentially expressed) and NANOG ChIP-seq data that we presented in the original version of the manuscript.

Consistent with other experiments, also based on the IF THE Patient-26 exhibits an even more extreme phenotype, exhibiting also aberrant OCT4 activity in addition to NANOG. This is consistent with the flowcytometry data (Fig. 2), which indicated the Patient-26 as having the largest population still double positive for the two pluripotency surface markers (SSEA4 and TRA-160) at the day-14.

7) SALL4 is strongly hypothesized by the authors to mediate the ARID1B repressive mechanism. There is not experimental evidence in the manuscript to support this. It should be omitted from the manuscript unless there is more supporting evidence.

As recommended by the Reviewer, we removed any reference to SALL4 from abstract, introduction and discussion (and supplementary Figure as well). We kept one small paragraph in the result section, since we think this is a potentially interesting finding, but we made sure to make no speculation about a potential contribution of SALL4 to the repressive regulatory activity of ARID1B-BAF at the pluripotency enhancers and promoters.

8) How does the author's model (Figure 7) fit in with published literature in BAF mutant animal models related to neural crest function?

The model was removed because of lack of space (a new figure was added with the new IF data). Nonetheless, we are happy to comment about this here. To our knowledge, there is only a BAF mutant mouse model with a conditional neural crest mutation, and it is specifically and ARID1A mutant (Chandler et al 2016). On the other hand, the ARID1B mouse models are not neural crest specific conditional KOs. Nonetheless, we looked for potential overlap between our set of 2,356 differentially expressed genes and a set of genes identified as differentially expressed in a recent RNA-seq study which compared cerebellar tissue from ARID1B^{+/-} and ARID1B-wt mice (Ellegood et al. 2021). Notably, 1,297 of the 2,356 genes (55%) identified as differentially expressed in our study were also found as

differentially expressed in the mouse model dataset. This overlap (55%) is significantly higher than expected by chance (Fisher's Exact Test $< 2.2 \times 10^{-16}$) and suggests that the pathways regulated by ARID1B are important for both craniofacial and brain development. This is not surprising, given that Coffin-Siris patients report both intellectual disabilities and craniofacial abnormalities.

Reviewer #2

In this current manuscript, Trizzino et al reprogrammed Coffin-Siris patient-derived skin fibroblasts into iPSCs and used it as a model system to study the function of ARID1B in directing cranial neural crest cell (CNCC) differentiation. They found that patient derived iPSCs showed impaired differentiation compared to control iPSC cell lines. Mechanistically, they found that due the haploinsufficiency of ARID1B, cells are unable to switch from ARID1A-BAF to ARID1B-BAF during induced differentiation, thus ARID1A-BAF remains at enhancers that regulating genes essential for stemness/pluripotency. Overall, the authors presented some interesting phenotypic discovery, while the mechanistic component lacks multiple critical controls. The enthusiasm is further reduced due to insufficient data analysis, and over-statement of some of the claims such as 'novel' ARID1B-BAF composition.

We thank the Reviewer for their kind words of appreciation of our study, and we apologize if specific controls were missing from some of the experiments. We have addressed this weakness in the revised version, by performing all of the experiments and analysis recommended by the Reviewer.

1) If a key point of the model (also indicated in the title) is to demonstrate the switch of ARID1A to ARID1B-BAF complex during differentiation, it would be essential to have ChIP-seq of ARID1A, ARID1B in iPSCs and differentiated cells (Day 5) both in control and patient cell lines, as well as ATAC-seq (and K27Ac, if the authors want to add the enhancer component story). Completion of these experiments will allow systematic cross comparison of ARID1A, ARID1B at stem cell states and CNCC states, correlate their targeting to chromatin accessibilities (and enhancers), as well as SOX2 and NANOG targeting. If the model stands, we would expect loss of ARID1A targeting (or altered), gained ARID1B targeting, correlations with changes on chromatin accessibilities, and how do they collectively regulate gene expression programs. The current data is not sufficient to support the model.

As recommended by the Reviewer, we have performed ChIP-seq for ARID1A and ARID1B, as well as ATAC-seq in all the four lines in both iPSCs and day-5. We also performed a H3K27ac time-course in the patients and control line. (see updated Figure 7 for ARID1A, Figure 4d and Supplementary Fig. 4; the figures are also reported at the end of this paragraph for Reviewer's convenience).

All these experiments combined supported our model. In fact, the 5,540 pluripotency enhancers are all bound by ARID1A, and not by ARID1B, at the iPSC stage in all of the four lines. This was expected, because all the known embryonic stem cell specific BAF configurations (esBAF, gBAF) incorporate ARID1A (esBAF) but do not incorporate ARID1B. At the day-5, in normal conditions (i.e. the two control lines) the 5,500 pluripotency enhancers lose ARID1A binding and gain ARID1B binding. Conversely, in the ARID1B-haplodeficient patient lines aberrant ARID1A binding is maintained at these enhancers, to compensate for the missing ARID1B.

ATAC-seq performed in iPSCs (day 0) revealed that the 5,540 regions were highly accessible, with no significant differences between patient and control lines (Supplementary Fig. S4a). By day 5, this dramatically shifted and 5,511 of the 5,540 regions (99.4%) were called as peaks exclusive to the patient lines. These data suggested that

these were regions highly accessible in iPSCs and repressed by day 5 of iPSC-to-CNCC differentiation. Such repression is impaired by ARID1B-haploinsufficiency in the patient lines, indicating that chromatin accessibility in the 5,540 genomic sites may be directly regulated by an ARID1B-containing BAF during exit from pluripotency and neuroectoderm specification.

2) Importantly, throughout the manuscript, critical control experiments are missing in multiple places: In Fig1. Should show similar staining on control iPSCs so that readers can compare the patient derived iPSCs (Fig1d, Fig1) to.

We thank the Reviewer for the suggestion. We have performed a new IF staining for the same markers on control iPSCs and in both Patients, and included it in updated Fig. 1d. These additional staining confirmed that at the iPSC stage there is no difference in pluripotency gene expression across lines.

3) In Fig3. Why align ARID1B peaks called in control lines to patient specific (gained) ATAC-seq peaks, when there is low ARID1B in patient-derived cell lines? Several controls are missing: ATAC-seq at iPSCs stage both in control and patient cells; ARID1B ChIP-seq in control iPSCs at Day 0 and Day 5/7. Ideally, one would compare ATAC-seq changes of Day 5 vs Day 0 in control cell line and patient cell lines; as well as compare control and patient cells at Day 0 and Day 5. In this case, the authors only compared Day 5 peaks between control and patient cell line.

We thank the Reviewer for pointing out that critical controls were missing. As answered in “Point 1” these additional experiments (ARID1A and ARID1B ChIP-seq and ATAC-seq in all the four lines at day-0 and day-5) were all performed. See answer to Point-1 for the details about the results.

4) Similarly, while it’s nice the authors performed time-course K27Ac ChIP-seq in the control cell line, it is critical to have that data in the patient cell lines. Do these regions maintain active during differentiation? Are there any de novo enhancers in the patient samples?

We thank the Reviewer for suggesting to perform a H3K27ac time-course in patients, which provided insightful results. These new experiments are reported in Fig 4g and also at the end of this paragraph for Reviewer’s convenience.

As mentioned, these new experiments suggested by the Reviewer provided important results. In fact, at the day-5, the majority of the enhancers still display H3K27ac signal also in the control line, although such signal is ~half than the one observed in the two patient-lines (Fig. 4g and see also below).

On the other hand, at the day-7, the picture is markedly different. Namely, at this time-point these enhancers show NO signal in the control, while still displaying strong signal in both patient lines. Finally, at the day-9 these enhancers resulted inactivated (i.e. no H3K27ac signal) also in Patient-19, while they are still fully active in Patient-26. This is consistent with other experiments (flowcytometry, immunofluorescence, NANOG ChIP-seq) which showed a more “extreme” molecular phenotype for the patient-26. In some ways, the patient-19 seems able to cope better than patient 26, but future studies, with a larger set of patients, will be needed to understand the patient-to-patient variability.

5) In Fig.4. Similarly, important controls are missing: RNA-seq of control and patient cell lines at Day 0, which will allow comparison/analysis of gene expression changes upon differentiation in the patient samples.

RNA-seq at day-0 was already included and discussed in the original version of the manuscript, we apologize if it was not sufficiently highlighted.

Consistent with the current knowledge that ARID1B is NOT a component of any of the BAF configurations known to be active at the iPSC stage (esBAF, gBAF), ARID1B-

haploinsufficiency results in only 57 differentially expressed genes at the iPSC stage (day-0), as opposed to 2,365 differentially expressed genes at the day-5 (time-point in which we report the peak of ARID1B protein activity).

6) In Fig6B: A time-course WB of ARID1A level in patient cells would be essential to assess how ARID1A protein level changes in the context of ARID1B haploinsufficiency.

Premise: Former Figure 6 is now Figure 7. We thank the Reviewer for this important suggestion. As recommended, we have performed the ARID1A time-course WB on both patient lines, and included it in the new Figure 7b, and also reported it at the end of this paragraph for Reviewer's convenience. Time-course immunoblotting confirmed that the temporary decommissioning of ARID1A during neuroectoderm specification (~days 1-7), which we previously observed in the control line (Fig. 7a), was NOT PERFORMED in neither of the patient lines (Fig. 7b). In fact, in both patient lines, ARID1A protein levels were consistent between the iPSCs and all the differentiation time points (Fig. 7b).

Together, these additional experiments confirmed our model, indicating that the patient cells compensated for the partial loss of ARID1B by maintaining aberrantly high ARID1A levels throughout the differentiation process.

7) In Fig6E: it is an over-statement of identifying a 'novel' ARID1B-BAF complex composition. Looking at the listed subunits, all of them are known BAF complex members, while SMARCA5 is NOT a SWI/SNF subunit.

The Reviewer is correct, all of the listed subunits are known BAF components. We referred to it as "novel" because this specific combination of subunits together was (to our knowledge) never described before. For instance, it is different from the npBAF and from the neuronal-BAF, both of which include ARID1B but with a different set of other BAF subunits. Nonetheless, we agree with the Reviewer that "novel" does not represent the most appropriate terminology. We therefore edited the language accordingly, replacing "novel" with "specific of the neural crest lineage". As suggested by the Reviewer, we also

removed any reference to SMARCA5, since it is not a BAF subunit. We had originally included it in the table because we found it interesting. In fact, SMARCA5 is part of ISWI, and to our knowledge this is the first time an ISWI subunit is associated with SWI/SNF. However, exploring possible associations between ISWI and SWI/SNF is out of the scopes of this paper, and future studies will be required to uncover the function of such interaction. Therefore, we agree with the reviewer that SMARCA5 should not be attributed to the ARID1B-BAF, nor mentioned in the manuscript.

8) While the authors claim only SMARCA4 is found in ARID1B IP, did they check the expression level of SMARCA2?

We apologize for the lack of clarity. SMARCA2 is expressed at the gene level, but is not incorporated into the ARID1B-BAF. We repeated two replicates of the IP/Mass-spec, and both times zero peptides of SMARCA2 coeluted with ARID1B, as opposed to >40 peptides of SMARCA4 which coeluted with ARID1B in both replicates (See updated Figure 7).

9) Another control would be ARID1A mass-spec.

As recommended by the Reviewer, we performed ARID1A IP/mass-spec at the day-5 on both patient-lines. The goal of this experiment was to investigate in the patient-lines if the aberrantly active ARID1A replaces ARID1B in the ARID1B-BAF or if instead it is part of a completely different BAF configuration. We thus performed ARID1A IP-MS at day-5 in both patient lines. Notably, in the two patient lines ARID1A co-eluted with all the other subunits of the ARID1B-BAF, suggesting that it DOES replace ARID1B in the ARID1B-BAF (See updated Supplementary Figure 5). We also performed the ARID1A mass-spec in the control line at day-5, to have a proper control. However, since very little ARID1A protein is detectable by WB in control conditions at this point (Fig. 7), the IP was unsuccessful.

Minor points:

Fig6A is a duplicate of Fig2c.

The legend of Fig 6A originally mentioned “duplicated from Fig. 2C for convenience”. Nonetheless, following the Reviewer’s suggestion, we have replaced the panel in Fig. 7A (Figure 6 is now Fig. 7) with a different replicate of the same experiment.

FigS2A: What is the value of Y-axis? Why is it drastically different from FigS2C?

We apologize for the missing axis label, which was supposed to read as “Mean Density/tag (50bp)”, exactly like in Fig. S2C.

Nonetheless, we have removed that panel (which only showed average profiles), and replaced it with heatmaps AND average profiles together (coming from new, higher quality ARID1B ChIP-seq replicates), which are now shown in the main Figure 4d.

FigS2B: Why would control shRNA have an effect on knocking down ARID1B expression?

We apologize if our quantitative western blot resulted confusing. The shRNA does not have an effect in knocking-down ARID1B expression. The knock-down is generated exclusively by the shARID1B.

More in detail, the figure displays a quantitative WB, with a gradient of protein lysate amount loaded in the gel. Therefore, as expected, 5ug of shCONTROL protein lysate generate a weaker ARID1B band in the WB than 20 ug of shCONTROL lysate.

We have now clarified this better in the figure legend.

FigS3: Should also show the level of ARID1B and adding IgG control

Following recommendation from another reviewer, the SALL4 data (including the co-IP) were removed from the paper.

Reviewer #3

Pagliaroli et al. show that iPSCs from Coffin-Siris patients with heterozygous ARID1B mutations fail to undergo an ARID1A-to-ARID1B switch at the onset of induced cranial neural crest differentiation, resulting in impaired lineage determination. The authors conclude that impaired cranial neural crest differentiation underlies Coffin-Siris pathogenesis. BAF complex switching during differentiation and SWI/SNF complex roles in both maintenance of and exit from pluripotency are known. The strengths of the manuscript lie in the use of patient-derived cell lines and robust genomic experiments. Although the studies are well-done and manuscript is well-written, the physiological relevance of the findings are still unclear, especially since all of the studies were performed in iPSCs and little effort is made to connect the findings back to the disease pathogenesis.

We thank the Reviewer for their kind words of appreciation of our study.

-Please increase use of statistical tests in the figures.

For every analysis and comparison that we present and discuss in the manuscript, we have performed accurate statistical testing, including multiple testing correction (FDR) on the genomic data when appropriate. Following Reviewer's suggestion, we now report all the p-values also in the figures, and test methodologies in the figure legends.

-Please include descriptions of the statistical test used in the figure legends.

Following Reviewer's suggestion, we included description of statistical tests in the legend.

-With only an N=2 of patients, any associations with disease severity and differentiation defect are too speculative.

We thank the Reviewer for pointing this out. We agree that with N=2, the association between differentiation defect and disease severity is too speculative. Therefore, following Reviewer's suggestion we removed the following sentence from the discussion: *"Nonetheless, these lines of evidence potentially establish a direct correlation between reduced attenuation of pluripotency enhancers, inefficient exit from pluripotency, impaired cell differentiation, and disease severity"*

-Previous studies have shown Coffin-Siris-like defects in ARID1B haploinsufficient mice. In addition, it's unclear if mouse models or Coffin-Siris patients have defects in early central nervous system or neural crest development, or both. Based on the major conclusion raised in manuscript-that cranial neural crest differentiation underlies Coffin-Siris pathogenesis-it seems important to address the cellular etiology of the disease or **at least determine if cranial neural crest differentiation genes identified are affected in vivo**. This would help determine if the disease is associated with cranial neural crest defects and thus further support the genomic mechanisms put forth in the manuscript.

We thank the Reviewer for rising this point. As recommended, we looked for overlap between our RNA-seq data and RNA-seq data generated in-vivo, in ARID1B^{+/-} mouse-model studies. In particular, there are two ARID1B mouse model studies that generated RNA-seq data: PMID 28695822 (RNA-seq from hippocampus) and PMID 33757588 (cerebellum).

The latter, was the only study which made the list of differentially expressed genes publicly available, and we thus we focused on these data.

Remarkably, 55% of the 2,365 genes identified as differentially expressed between patient and controls in our study (day-5 of iPSC-to-CNCC differentiation) were also found as differentially expressed in the cerebellum of ARID1B^{+/-} mice compared to ARID1B-wt mice. This overlap is significantly higher than expected by chance (Fisher's Exact Test $p < 2.2 \times 10^{-16}$) and suggests that the genes we identified as differentially expressed "in vitro" (i.e. in differentiating iPSCs) are affected also in vivo. It is worth noting that the day-5 of iPSC-to-CNCC differentiation (time-point in which ARID1B is the most active) coincides with a developmental stage in which the iPSCs differentiate into neuroectodermal spheres. Once the neuroectodermal spheres are formed, they attach to the plate (~usually between day-6 and day-8) and start differentiating into cranial neural crest cells (days 9-14). During this latter developmental stage ARID1B is decommissioned, and ARID1A re-activated. Overall this suggests that ARID1B is crucial for exit from pluripotency and for neuroectoderm specification, why ARID1A regulates the developmental transition from neuroectoderm to neural crest.

Importantly, the neuroectoderm not only gives origin to the neural crest cells, but it is also a precursor of central nervous system (forebrain, midbrain, hindbrain and motor neurons). Therefore, given the overlap of genes differentially expressed in both the ARID1B^{+/-} neural crest and in the ARID1B^{+/-} mouse cerebellum, one could speculate that the dysregulation of neuroectoderm specification caused by *ARID1B* mutations may underlie both the craniofacial abnormalities and the cognitive impairment typical of the Coffin-Siris syndrome. We are discussing this point in the updated discussion.

REVIEWERS' COMMENTS

Reviewer #1 (Remarks to the Author):

In the resubmission of this manuscript, the authors have addressed the concerns in my initial review and have thoroughly strengthened the manuscript.

Reviewer #2 (Remarks to the Author):

Revision

In this revised version, the authors performed additional experiments suggested by all three reviewers. While the additional data significantly improved the work, it is convincing that deficiency of ARID1B is potentially an underlying mechanism of the phenotype they observed, there are still a few issues remain:

1. There is little or no evidence supporting the repressive role of ARID1B in regulating chromatin accessibility. Rather, the 'patient specific open chromatin regions' are bound by ARID1A, which could be the reason why these regions remain open. In other words, loss of these regions in control/WT cells upon differentiation is potentially due to loss of ARID1A targeting (Figure 4d, 7d).
2. Related to this, while the reviewer appreciates the new time-course K27Ac, new ARID1A ChIP-seq experiments, the lack of basic stats/QC of these NGS data is disappointing. Readers would like to have a general idea of how the overall targeting of ARID1A, ARID1B changes. The author's rationale is to focus on patient specific ATAC regions throughout the analysis, but standard stats showing total number of K27Ac, ARID1A, ARID1B peaks, and heatmaps should be included. Similarly, are there additional ARID1A bound regions in the patients upon differentiation beyond these patient ATAC peaks? If so, what are the features of these regions? Without these stats, it is very difficult to assess the quality of these NGS data. In addition, a few representative IGV samples should be added to show binding of ARID1A, ARID1B, K27Ac, which will further help assess the ChIP-seq data quality.
3. Figure 2c and Figure 7a where the authors repeated this WB, but it seems like there are quite some variations regarding the expression of ARID1B. This could be either biological or technical issues. In either case, ARID1B level is not low as the authors stated in the iPSC stage. Also, the author cited a paper (PMID: 19279220) claiming ARID1B is not present in esBAF is not accurate, as they did show ARID1B in their data about esBAF.
4. As the authors also noted in the manuscript, with only 2 patient lines, there are already some substantial variations, caution needs to be added to interpreting some of the conclusions, especially how broadly the mechanism could be.

Reviewer #3 (Remarks to the Author):

The authors have addressed my concerns.

1. There is little or no evidence supporting the repressive role of ARID1B in regulating chromatin accessibility. Rather, the 'patient specific open chromatin regions' are bound by ARID1A, which could be the reason why these regions remain open. In other words, loss of these regions in control/WT cells upon differentiation is potentially due to loss of ARID1A targeting (Figure 4d, 7d).

We appreciate this comment from the reviewer. To address it, we added the following sentence to the discussion: “It is worth noting that future studies will be necessary to determine if enhancer repression is directly performed by ARID1B, or if ARID1B role is rather to prevent ARID1A from binding at these enhancers to keep them active”.

2. Related to this, while the reviewer appreciates the new time-course K27Ac, new ARID1A ChIP-seq experiments, the lack of basic stats/QC of these NGS data is disappointing. Readers would like to have a general idea of how the overall targeting of ARID1A, ARID1B changes. The author's rationale is to focus on patient specific ATAC regions throughout the analysis, but standard stats showing total number of K27Ac, ARID1A, ARID1B peaks, and heatmaps should be included. Similarly, are there additional ARID1A bound regions in the patients upon differentiation beyond these patient ATAC peaks? If so, what are the features of these regions? Without these stats, it is very difficult to assess the quality of these NGS data. In addition, a few representative IGV samples should be added to show binding of ARID1A, ARID1B, K27Ac, which will further help assess the ChIP-seq data quality.

As recommended by the Reviewer, we added genome browser screenshots to figures 4 and 7, and also added the peak numbers in the manuscript. We also added a sentence to comment that based on peak numbers, the aberrantly active ARID1A at day-5 seems to have an activity which goes beyond the patient-specific ATAC-seq peaks in at least one of the patients. The peak numbers for all the experiments are also included in the uploaded reporting summary, which is part of the supplementary materials.

3. Figure 2c and Figure 7a where the authors repeated this WB, but it seems like there are quite some variations regarding the expression of ARID1B. This could be either biological or technical issues. In either case, ARID1B level is not low as the authors stated in the iPSC stage. Also, the author cited a paper (PMID: 19279220) claiming ARID1B is not present in esBAF is not accurate, as they did show ARID1B in their data about esBAF.

We believe that the differences between the two replicates are biological, because ARID1B's inactivation it's part of a highly dynamic process takes place when the floating neuroectodermal spheres attach to the plate and start differentiating into CNCCs. Regardless of such biological variation, the key stages when ARID1B's role is important for CNCCs formation (days 1-5 or as written in the paper “the window of robust ARID1B expression”) are consistent between the blots

Regarding the low ARID1B level at iPSC stage, that sentence was already removed in the previously revised manuscript. Finally, regarding ARID1B as a member of the esBAF, while it is true that it can be incorporated, the authors of that study quantified ARID1B's relative abundance in the complex as near zero, in contrast to a very abundant ARID1A. We thus

made the following change in the introduction: *"Importantly, the esBAF exclusively incorporates ARID1A and not ARID1B"* changed with *"Importantly, the esBAF predominantly incorporates ARID1A and very rarely ARID1B (relative ARID1B abundance in the esBAF was quantified as ~0 by Ho et al.)"*

4. As the authors also noted in the manuscript, with only 2 patient lines, there are already some substantial variations, caution needs to be added to interpreting some of the conclusions, especially how broadly the mechanism could be.

We appreciate this comment from the reviewer. To address it, we added the following sentence to the discussion: *"Even if our study is based on two patient vs two control lines, multiple assays (ATAC-seq, ChIP-seq, RNA-seq and differentiation assays) highlighted important differences during the differentiation of wild-type vs ARID1B-mutant iPSCs to CNCCs. However, additional experiments with a larger set of patient-derived cell lines would be required to support this model."*